# Multi-Class Deep Boosting

**Vitaly Kuznetsov**
Courant Institute
251 Mercer Street
New York, NY 10012
vitaly@cims.nyu.edu

**Mehryar Mohri**
Courant Institute & Google Research
251 Mercer Street
New York, NY 10012
mohri@cims.nyu.edu

**Umar Syed**
Google Research
76 Ninth Avenue
New York, NY 10011
usyed@google.com

## Abstract

We present new ensemble learning algorithms for multi-class classification. Our algorithms can use as a base classifier set a family of deep decision trees or other rich or complex families and yet benefit from strong generalization guarantees. We give new data-dependent learning bounds for convex ensembles in the multi-class classification setting expressed in terms of the Rademacher complexities of the sub-families composing the base classifier set, and the mixture weight assigned to each sub-family. These bounds are finer than existing ones both thanks to an improved dependency on the number of classes and, more crucially, by virtue of a more favorable complexity term expressed as an average of the Rademacher complexities based on the ensemble's mixture weights. We introduce and discuss several new multi-class ensemble algorithms benefiting from these guarantees, prove positive results for the $H$-consistency of several of them, and report the results of experiments showing that their performance compares favorably with that of multi-class versions of AdaBoost and Logistic Regression and their $L_1$-regularized counterparts.

## 1 Introduction

Devising ensembles of base predictors is a standard approach in machine learning which often helps improve performance in practice. Ensemble methods include the family of boosting meta-algorithms among which the most notable and widely used one is AdaBoost [Freund and Schapire, 1997], also known as forward stagewise additive modeling [Friedman et al., 1998]. AdaBoost and its other variants learn convex combinations of predictors. They seek to greedily minimize a convex surrogate function upper bounding the misclassification loss by augmenting, at each iteration, the current ensemble, with a new suitably weighted predictor.

One key advantage of AdaBoost is that, since it is based on a stagewise procedure, it can learn an effective ensemble of base predictors chosen from a very large and potentially infinite family, provided that an efficient algorithm is available for selecting a good predictor at each stage. Furthermore, AdaBoost and its $L_1$-regularized counterpart [Rätsch et al., 2001a] benefit from favorable learning guarantees, in particular theoretical margin bounds [Schapire et al., 1997, Koltchinskii and Panchenko, 2002]. However, those bounds depend not just on the margin and the sample size, but also on the complexity of the base hypothesis set, which suggests a risk of overfitting when using too complex base hypothesis sets. And indeed, overfitting has been reported in practice for AdaBoost in the past [Grove and Schuurmans, 1998, Schapire, 1999, Dietterich, 2000, Rätsch et al., 2001b].

Cortes, Mohri, and Syed [2014] introduced a new ensemble algorithm, *DeepBoost*, which they proved to benefit from finer learning guarantees, including favorable ones even when using as base classifier set relatively rich families, for example a family of very deep decision trees, or other similarly complex families. In DeepBoost, the decisions in each iteration of which classifier to add to the ensemble and which weight to assign to that classifier, depend on the (data-dependent) complexity

of the sub-family to which the classifier belongs – one interpretation of DeepBoost is that it applies the principle of structural risk minimization to each iteration of boosting. Cortes, Mohri, and Syed [2014] further showed that empirically DeepBoost achieves a better performance than AdaBoost, Logistic Regression, and their $L_1$-regularized variants. The main contribution of this paper is an extension of these theoretical, algorithmic, and empirical results to the multi-class setting.

Two distinct approaches have been considered in the past for the definition and the design of boosting algorithms in the multi-class setting. One approach consists of combining base classifiers mapping each example $x$ to an output label $y$. This includes the SAMME algorithm [Zhu et al., 2009] as well as the algorithm of Mukherjee and Schapire [2013], which is shown to be, in a certain sense, optimal for this approach. An alternative approach, often more flexible and more widely used in applications, consists of combining base classifiers mapping each pair $(x, y)$ formed by an example $x$ and a label $y$ to a real-valued score. This is the approach adopted in this paper, which is also the one used for the design of AdaBoost.MR [Schapire and Singer, 1999] and other variants of that algorithm.

In Section 2, we prove a novel generalization bound for multi-class classification ensembles that depends only on the Rademacher complexity of the hypothesis classes to which the classifiers in the ensemble belong. Our result generalizes the main result of Cortes et al. [2014] to the multi-class setting, and also represents an improvement on the multi-class generalization bound due to Koltchinskii and Panchenko [2002], even if we disregard our finer analysis related to Rademacher complexity. In Section 3, we present several multi-class surrogate losses that are motivated by our generalization bound, and discuss and compare their functional and consistency properties. In particular, we prove that our surrogate losses are *realizable H-consistent*, a hypothesis-set-specific notion of consistency that was recently introduced by Long and Servedio [2013]. Our results generalize those of Long and Servedio [2013] and admit simpler proofs. We also present a family of multi-class DeepBoost learning algorithms based on each of these surrogate losses, and prove general convergence guarantee for them. In Section 4, we report the results of experiments demonstrating that multi-class DeepBoost outperforms AdaBoost.MR and multinomial (additive) logistic regression, as well as their $L_1$-norm regularized variants, on several datasets.

## 2   Multi-class data-dependent learning guarantee for convex ensembles

In this section, we present a data-dependent learning bound in the multi-class setting for convex ensembles based on multiple base hypothesis sets. Let $\mathcal{X}$ denote the input space. We denote by $\mathcal{Y} = \{1, \ldots, c\}$ a set of $c \geq 2$ classes. The label associated by a hypothesis $f \colon \mathcal{X} \times \mathcal{Y} \to \mathbb{R}$ to $x \in \mathcal{X}$ is given by $\operatorname{argmax}_{y \in \mathcal{Y}} f(x, y)$. The margin $\rho_f(x, y)$ of the function $f$ for a labeled example $(x, y) \in \mathcal{X} \times \mathcal{Y}$ is defined by

$$\rho_f(x, y) = f(x, y) - \max_{y' \neq y} f(x, y'). \tag{1}$$

Thus, $f$ misclassifies $(x, y)$ iff $\rho_f(x, y) \leq 0$. We consider $p$ families $H_1, \ldots, H_p$ of functions mapping from $\mathcal{X} \times \mathcal{Y}$ to $[0, 1]$ and the ensemble family $\mathcal{F} = \operatorname{conv}(\bigcup_{k=1}^p H_k)$, that is the family of functions $f$ of the form $f = \sum_{t=1}^T \alpha_t h_t$, where $\boldsymbol{\alpha} = (\alpha_1, \ldots, \alpha_T)$ is in the simplex $\Delta$ and where, for each $t \in [1, T]$, $h_t$ is in $H_{k_t}$ for some $k_t \in [1, p]$. We assume that training and test points are drawn i.i.d. according to some distribution $\mathcal{D}$ over $\mathcal{X} \times \mathcal{Y}$ and denote by $S = ((x_1, y_1), \ldots, (x_m, y_m))$ a training sample of size $m$ drawn according to $\mathcal{D}^m$. For any $\rho > 0$, the generalization error $R(f)$, its $\rho$-margin error $R_\rho(f)$ and its empirical margin error are defined as follows:

$$R(f) = \operatorname*{E}_{(x,y) \sim \mathcal{D}}[1_{\rho_f(x,y) \leq 0}], \quad R_\rho(f) = \operatorname*{E}_{(x,y) \sim \mathcal{D}}[1_{\rho_f(x,y) \leq \rho}], \text{ and } \widehat{R}_{S,\rho}(f) = \operatorname*{E}_{(x,y) \sim S}[1_{\rho_f(x,y) \leq \rho}], \tag{2}$$

where the notation $(x, y) \sim S$ indicates that $(x, y)$ is drawn according to the empirical distribution defined by $S$. For any family of hypotheses $G$ mapping $\mathcal{X} \times \mathcal{Y}$ to $\mathbb{R}$, we define $\Pi_1(G)$ by

$$\Pi_1(G) = \{x \mapsto h(x, y) \colon y \in \mathcal{Y}, h \in G\}. \tag{3}$$

The following theorem gives a margin-based Rademacher complexity bound for learning with ensembles of base classifiers with multiple hypothesis sets. As with other Rademacher complexity learning guarantees, our bound is data-dependent, which is an important and favorable characteristic of our results.

**Theorem 1.** *Assume $p > 1$ and let $H_1, \ldots, H_p$ be $p$ families of functions mapping from $\mathcal{X} \times \mathcal{Y}$ to $[0,1]$. Fix $\rho > 0$. Then, for any $\delta > 0$, with probability at least $1 - \delta$ over the choice of a sample $S$ of size $m$ drawn i.i.d. according to $\mathcal{D}$, the following inequality holds for all $f = \sum_{t=1}^{T} \alpha_t h_t \in \mathcal{F}$:*

$$R(f) \leq \widehat{R}_{S,\rho}(f) + \frac{8c}{\rho} \sum_{t=1}^{T} \alpha_t \mathfrak{R}_m(\Pi_1(H_{k_t})) + \frac{2}{c\rho} \sqrt{\frac{\log p}{m}} + \sqrt{\left\lceil \frac{4}{\rho^2} \log\left(\frac{c^2 \rho^2 m}{4 \log p}\right)\right\rceil \frac{\log p}{m} + \frac{\log \frac{2}{\delta}}{2m}},$$

*Thus, $R(f) \leq \widehat{R}_{S,\rho}(f) + \frac{8c}{\rho} \sum_{t=1}^{T} \alpha_t \mathfrak{R}_m(H_{k_t}) + O\left(\sqrt{\frac{\log p}{\rho^2 m}} \log\left[\frac{\rho^2 c^2 m}{4 \log p}\right]\right)$.*

The full proof of theorem 3 is given in Appendix B. Even for $p = 1$, that is for the special case of a single hypothesis set, our analysis improves upon the multi-class margin bound of Koltchinskii and Panchenko [2002] since our bound admits only a linear dependency on the number of classes $c$ instead of a quadratic one. However, the main remarkable benefit of this learning bound is that its complexity term admits an explicit dependency on the mixture coefficients $\alpha_t$. It is a weighted average of Rademacher complexities with mixture weights $\alpha_t$, $t \in [1, T]$. Thus, the second term of the bound suggests that, while some hypothesis sets $H_k$ used for learning could have a large Rademacher complexity, this may not negatively affect generalization if the corresponding total mixture weight (sum of $\alpha_t$s corresponding to that hypothesis set) is relatively small. Using such potentially complex families could help achieve a better margin on the training sample.

The theorem cannot be proven via the standard Rademacher complexity analysis of Koltchinskii and Panchenko [2002] since the complexity term of the bound would then be $\mathfrak{R}_m(\text{conv}(\bigcup_{k=1}^{p} H_k)) = \mathfrak{R}_m(\bigcup_{k=1}^{p} H_k)$ which does not admit an explicit dependency on the mixture weights and is lower bounded by $\sum_{t=1}^{T} \alpha_t \mathfrak{R}_m(H_{k_t})$. Thus, the theorem provides a finer learning bound than the one obtained via a standard Rademacher complexity analysis.

## 3 Algorithms

In this section, we will use the learning guarantees just described to derive several new ensemble algorithms for multi-class classification.

### 3.1 Optimization problem

Let $H_1, \ldots, H_p$ be $p$ disjoint families of functions taking values in $[0,1]$ with increasing Rademacher complexities $\mathfrak{R}_m(H_k)$, $k \in [1, p]$. For any hypothesis $h \in \cup_{k=1}^{p} H_k$, we denote by $d(h)$ the index of the hypothesis set it belongs to, that is $h \in H_{d(h)}$. The bound of Theorem 3 holds uniformly for all $\rho > 0$ and functions $f \in \text{conv}(\bigcup_{k=1}^{p} H_k)$. Since the last term of the bound does not depend on $\boldsymbol{\alpha}$, it suggests selecting $\boldsymbol{\alpha}$ that would minimize:

$$G(\boldsymbol{\alpha}) = \frac{1}{m} \sum_{i=1}^{m} 1_{\rho_f(x_i, y_i) \leq \rho} + \frac{8c}{\rho} \sum_{t=1}^{T} \alpha_t r_t,$$

where $r_t = \mathfrak{R}_m(H_{d(h_t)})$ and $\boldsymbol{\alpha} \in \Delta$.[1] Since for any $\rho > 0$, $f$ and $f/\rho$ admit the same generalization error, we can instead search for $\alpha \geq 0$ with $\sum_{t=1}^{T} \alpha_t \leq 1/\rho$, which leads to

$$\min_{\boldsymbol{\alpha} \geq 0} \frac{1}{m} \sum_{i=1}^{m} 1_{\rho_f(x_i, y_i) \leq 1} + 8c \sum_{t=1}^{T} \alpha_t r_t \quad \text{s.t.} \sum_{t=1}^{T} \alpha_t \leq \frac{1}{\rho}. \tag{4}$$

The first term of the objective is not a convex function of $\boldsymbol{\alpha}$ and its minimization is known to be computationally hard. Thus, we will consider instead a convex upper bound. Let $u \mapsto \Phi(-u)$ be a non-increasing convex function upper-bounding $u \mapsto 1_{u \leq 0}$ over $\mathbb{R}$. $\Phi$ may be selected to be

for example the exponential function as in AdaBoost [Freund and Schapire, 1997] or the logistic function. Using such an upper bound, we obtain the following convex optimization problem:

$$\min_{\boldsymbol{\alpha} \geq 0} \frac{1}{m} \sum_{i=1}^{m} \Phi\Big(1 - \rho_f(x_i, y_i)\Big) + \lambda \sum_{t=1}^{T} \alpha_t r_t \quad \text{s.t.} \sum_{t=1}^{T} \alpha_t \leq \frac{1}{\rho}, \tag{5}$$

where we introduced a parameter $\lambda \geq 0$ controlling the balance between the magnitude of the values taken by function $\Phi$ and the second term.[2] Introducing a Lagrange variable $\beta \geq 0$ associated to the constraint in (5), the problem can be equivalently written as

$$\min_{\boldsymbol{\alpha} \geq 0} \frac{1}{m} \sum_{i=1}^{m} \Phi\Big(1 - \min_{y \neq y_i}\Big[ \sum_{t=1}^{T} \alpha_t h_t(x_i, y_i) - \alpha_t h_t(x_i, y)\Big]\Big) + \sum_{t=1}^{T}(\lambda r_t + \beta)\alpha_t.$$

Here, $\beta$ is a parameter that can be freely selected by the algorithm since any choice of its value is equivalent to a choice of $\rho$ in (5). Since $\Phi$ is a non-decreasing function, the problem can be equivalently written as

$$\min_{\boldsymbol{\alpha} \geq 0} \frac{1}{m} \sum_{i=1}^{m} \max_{y \neq y_i} \Phi\Big(1 - \Big[ \sum_{t=1}^{T} \alpha_t h_t(x_i, y_i) - \alpha_t h_t(x_i, y)\Big]\Big) + \sum_{t=1}^{T}(\lambda r_t + \beta)\alpha_t.$$

Let $\{h_1, \ldots, h_N\}$ be the set of distinct base functions, and let $F_{\max}$ be the objective function based on that expression:

$$F_{\max}(\boldsymbol{\alpha}) = \frac{1}{m} \sum_{i=1}^{m} \max_{y \neq y_i} \Phi\Big(1 - \sum_{j=1}^{N} \alpha_j \mathsf{h}_j(x_i, y_i, y)\Big) + \sum_{j=1}^{N} \Lambda_j \alpha_j, \tag{6}$$

with $\boldsymbol{\alpha} = (\alpha_1, \ldots, \alpha_N) \in \mathbb{R}^N$, $\mathsf{h}_j(x_i, y_i, y) = h_j(x_i, y_i) - h_j(x_i, y)$, and $\Lambda_j = \lambda r_j + \beta$ for all $j \in [1, N]$. Then, our optimization problem can be rewritten as $\min_{\boldsymbol{\alpha} \geq 0} F_{\max}(\boldsymbol{\alpha})$. This defines a convex optimization problem since the domain $\{\boldsymbol{\alpha} \geq 0\}$ is a convex set and since $F_{\max}$ is convex: each term of the sum in its definition is convex as a pointwise maximum of convex functions (composition of the convex function $\Phi$ with an affine function) and the second term is a linear function of $\boldsymbol{\alpha}$. In general, $F_{\max}$ is not differentiable even when $\Phi$ is, but, since it is convex, it admits a sub-differential at every point. Additionally, along each direction, $F_{\max}$ admits left and right derivatives both non-increasing and a differential everywhere except for a set that is at most countable.

## 3.2 Alternative objective functions

We now consider the following three natural upper bounds on $F_{\max}$ which admit useful properties that we will discuss later, the third one valid when $\Phi$ can be written as the composition of two function $\Phi_1$ and $\Phi_2$ with $\Phi_1$ a non-increasing function:

$$F_{\text{sum}}(\boldsymbol{\alpha}) = \frac{1}{m} \sum_{i=1}^{m} \sum_{y \neq y_i} \Phi\Big(1 - \sum_{j=1}^{N} \alpha_j \mathsf{h}_j(x_i, y_i, y)\Big) + \sum_{j=1}^{N} \Lambda_j \alpha_j \tag{7}$$

$$F_{\text{maxsum}}(\boldsymbol{\alpha}) = \frac{1}{m} \sum_{i=1}^{m} \Phi\Big(1 - \sum_{j=1}^{N} \alpha_j \rho_{h_j}(x_i, y_i)\Big) + \sum_{j=1}^{N} \Lambda_j \alpha_j \tag{8}$$

$$F_{\text{compsum}}(\boldsymbol{\alpha}) = \frac{1}{m} \sum_{i=1}^{m} \Phi_1\Big( \sum_{y \neq y_i} \Phi_2\Big(1 - \sum_{j=1}^{N} \alpha_j \mathsf{h}_j(x_i, y_i, y)\Big)\Big) + \sum_{j=1}^{N} \Lambda_j \alpha_j. \tag{9}$$

$F_{\text{sum}}$ is obtained from $F_{\max}$ simply by replacing in the definition of $F_{\max}$ the max operator by a sum. Clearly, function $F_{\text{sum}}$ is convex and inherits the differentiability properties of $\Phi$. A drawback of $F_{\text{sum}}$ is that for problems with very large $c$ as in structured prediction, the computation of the sum

may require resorting to approximations. $F_{\max\text{sum}}$ is obtained from $F_{\max}$ by noticing that, by the sub-additivity of the $\max$ operator, the following inequality holds:

$$\max_{y \neq y_i} \sum_{j=1}^{N} -\alpha_j \mathsf{h}_j(x_i, y_i, y) \leq \sum_{j=1}^{N} \max_{y \neq y_i} -\alpha_j \mathsf{h}_j(x_i, y_i, y) = \sum_{j=1}^{N} \alpha_j \rho_{h_j}(x_i, y_i).$$

As with $F_{\text{sum}}$, function $F_{\max\text{sum}}$ is convex and admits the same differentiability properties as $\Phi$. Unlike $F_{\text{sum}}$, $F_{\max\text{sum}}$ does not require computing a sum over the classes. Furthermore, note that the expressions $\rho_{h_j}(x_i, y_i)$, $i \in [1, m]$, can be pre-computed prior to the application of any optimization algorithm. Finally, for $\Phi = \Phi_1 \circ \Phi_2$ with $\Phi_1$ non-increasing, the $\max$ operator can be replaced by a sum before applying $\phi_1$, as follows:

$$\max_{y \neq y_i} \Phi\Big(1 - \mathsf{f}(x_i, y_i, y)\Big) = \Phi_1\Big(\max_{y \neq y_i} \Phi_2\big(1 - \mathsf{f}(x_i, y_i, y)\big)\Big) \leq \Phi_1\Big(\sum_{y \neq y_i} \Phi_2\big(1 - \mathsf{f}(x_i, y_i, y)\big)\Big),$$

where $\mathsf{f}(x_i, y_i, y) = \sum_{j=1}^{N} \alpha_j \mathsf{h}_j(x_i, y_i, y)$. This leads to the definition of $F_{\text{compsum}}$.

In Appendix C, we discuss the consistency properties of the loss functions just introduced. In particular, we prove that the loss functions associated to $F_{\max}$ and $F_{\text{sum}}$ are *realizable H-consistent* (see Long and Servedio [2013]) in the common cases where the exponential or logistic losses are used and that, similarly, in the common case where $\Phi_1(u) = \log(1 + u)$ and $\Phi_2(u) = \exp(u + 1)$, the loss function associated to $F_{\text{compsum}}$ is $H$-consistent.

Furthermore, in Appendix D, we show that, under some mild assumptions, the objective functions we just discussed are essentially within a constant factor of each other. Moreover, in the case of binary classification all of these objectives coincide.

### 3.3 Multi-class DeepBoost algorithms

In this section, we discuss in detail a family of multi-class DeepBoost algorithms, which are derived by application of coordinate descent to the objective functions discussed in the previous paragraphs. We will assume that $\Phi$ is differentiable over $\mathbb{R}$ and that $\Phi'(u) \neq 0$ for all $u$. This condition is not necessary, in particular, our presentation can be extended to non-differentiable functions such as the hinge loss, but it simplifies the presentation. In the case of the objective function $F_{\max\text{sum}}$, we will assume that both $\Phi_1$ and $\Phi_2$, where $\Phi = \Phi_1 \circ \Phi_2$, are differentiable. Under these assumptions, $F_{\text{sum}}$, $F_{\max\text{sum}}$, and $F_{\text{compsum}}$ are differentiable. $F_{\max}$ is not differentiable due to the presence of the $\max$ operators in its definition, but it admits a sub-differential at every point.

For convenience, let $\boldsymbol{\alpha}_t = (\alpha_{t,1}, \ldots, \alpha_{t,N})^\top$ denote the vector obtained after $t \geq 1$ iterations and let $\boldsymbol{\alpha}_0 = \mathbf{0}$. Let $\mathbf{e}_k$ denote the $k$th unit vector in $\mathbb{R}^N$, $k \in [1, N]$. For a differentiable objective $F$, we denote by $F'(\boldsymbol{\alpha}, \mathbf{e}_j)$ the directional derivative of $F$ along the direction $\mathbf{e}_j$ at $\boldsymbol{\alpha}$. Our coordinate descent algorithm consists of first determining the direction of maximal descent, that is $k = \operatorname{argmax}_{j \in [1, N]} |F'(\boldsymbol{\alpha}_{t-1}, \mathbf{e}_j)|$, next of determining the best step $\eta$ along that direction that preserves non-negativity of $\alpha$, $\eta = \operatorname{argmin}_{\boldsymbol{\alpha}_{t-1} + \eta \mathbf{e}_k \geq 0} F(\boldsymbol{\alpha}_{t-1} + \eta \mathbf{e}_k)$, and updating $\boldsymbol{\alpha}_{t-1}$ to $\boldsymbol{\alpha}_t = \boldsymbol{\alpha}_{t-1} + \eta \mathbf{e}_k$. We will refer to this method as *projected coordinate descent*. The following theorem provides a convergence guarantee for our algorithms in that case.

**Theorem 2.** *Assume that $\Phi$ is twice differentiable and that $\Phi''(u) > 0$ for all $u \in \mathbb{R}$. Then, the projected coordinate descent algorithm applied to $F$ converges to the solution $\boldsymbol{\alpha}^*$ of the optimization $\max_{\boldsymbol{\alpha} \geq 0} F(\boldsymbol{\alpha})$ for $F = F_{sum}$, $F = F_{maxsum}$, or $F = F_{compsum}$. If additionally $\Phi$ is strongly convex over the path of the iterates $\boldsymbol{\alpha}_t$, then there exists $\tau > 0$ and $\gamma > 0$ such that for all $t > \tau$,*

$$F(\boldsymbol{\alpha}_{t+1}) - F(\boldsymbol{\alpha}^*) \leq (1 - \tfrac{1}{\gamma})(F(\boldsymbol{\alpha}_t) - F(\boldsymbol{\alpha}^*)). \tag{10}$$

The proof is given in Appendix I and is based on the results of Luo and Tseng [1992]. The theorem can in fact be extended to the case where instead of the best direction, the derivative for the direction selected at each round is within a constant threshold of the best [Luo and Tseng, 1992]. The conditions of Theorem 2 hold for many cases in practice, in particular in the case of the exponential loss ($\Phi = \exp$) or the logistic loss ($\Phi(-x) = \log_2(1 + e^{-x})$). In particular, linear convergence is guaranteed in those cases since both the exponential and logistic losses are strongly convex over a compact set containing the converging sequence of $\boldsymbol{\alpha}_t$s.

MDEEPBOOSTSUM$(S = ((x_1, y_1), \ldots, (x_m, y_m)))$

1    **for** $i \leftarrow 1$ **to** $m$ **do**
2      **for** $y \in \mathcal{Y} - \{y_i\}$ **do**
3        $\mathcal{D}_1(i, y) \leftarrow \frac{1}{m(c-1)}$
4    **for** $t \leftarrow 1$ **to** $T$ **do**
5      $k \leftarrow \underset{j \in [1,N]}{\operatorname{argmin}} \; \epsilon_{t,j} + \frac{\Lambda_j m}{2 S_t}$
6      **if** $\left( (1 - \epsilon_{t,k}) e^{\alpha_{t-1,k}} - \epsilon_{t,k} e^{-\alpha_{t-1,k}} < \frac{\Lambda_k m}{S_t} \right)$ **then**
7        $\eta_t \leftarrow -\alpha_{t-1,k}$
8      **else** $\eta_t \leftarrow \log \left[ -\frac{\Lambda_k m}{2 \epsilon_t S_t} + \sqrt{\left[\frac{\Lambda_k m}{2 \epsilon_t S_t}\right]^2 + \frac{1 - \epsilon_t}{\epsilon_t}} \right]$
9      $\boldsymbol{\alpha}_t \leftarrow \boldsymbol{\alpha}_{t-1} + \eta_t \mathbf{e}_k$
10     $S_{t+1} \leftarrow \sum_{i=1}^m \sum_{y \neq y_i} \Phi'\big(1 - \sum_{j=1}^N \alpha_{t,j} \mathsf{h}_j(x_i, y_i, y)\big)$
11     **for** $i \leftarrow 1$ **to** $m$ **do**
12       **for** $y \in \mathcal{Y} - \{y_i\}$ **do**
13         $\mathcal{D}_{t+1}(i, y) \leftarrow \frac{\Phi'\big(1 - \sum_{j=1}^N \alpha_{t,j} \mathsf{h}_j(x_i, y_i, y)\big)}{S_{t+1}}$
14   $f \leftarrow \sum_{j=1}^N \alpha_{t,j} \mathsf{h}_j$
15   **return** $f$

Figure 1: Pseudocode of the MDeepBoostSum algorithm for both the exponential loss and the logistic loss. The expression of the weighted error $\epsilon_{t,j}$ is given in (12).

We will refer to the algorithm defined by projected coordinate descent applied to $F_{\text{sum}}$ by MDeepBoostSum, to $F_{\text{maxsum}}$ by MDeepBoostMaxSum, to $F_{\text{compsum}}$ by MDeepBoostCompSum, and to $F_{\text{max}}$ by MDeepBoostMax. In the following, we briefly describe MDeepBoostSum, including its pseudocode. We give a detailed description of all of these algorithms in the supplementary material: MDeepBoostSum (Appendix E), MDeepBoostMaxSum (Appendix F), MDeepBoostCompSum (Appendix G), MDeepBoostMax (Appendix H).

Define $\mathsf{f}_{t-1} = \sum_{j=1}^N \alpha_{t-1,j} \mathsf{h}_j$. Then, $F_{\text{sum}}(\boldsymbol{\alpha}_{t-1})$ can be rewritten as follows:

$$F_{\text{sum}}(\boldsymbol{\alpha}_{t-1}) = \frac{1}{m} \sum_{i=1}^m \sum_{y \neq y_i} \Phi\Big(1 - \mathsf{f}_{t-1}(x_i, y_i, y)\Big) + \sum_{j=1}^N \Lambda_j \alpha_{t-1,j}.$$

For any $t \in [1, T]$, we denote by $\mathcal{D}_t$ the distribution over $[1, m] \times [1, c]$ defined for all $i \in [1, m]$ and $y \in \mathcal{Y} - \{y_i\}$ by

$$\mathcal{D}_t(i, y) = \frac{\Phi'\big(1 - \mathsf{f}_{t-1}(x_i, y_i, y)\big)}{S_t}, \tag{11}$$

where $S_t$ is a normalization factor, $S_t = \sum_{i=1}^m \sum_{y \neq y_i} \Phi'(1 - \mathsf{f}_{t-1}(x_i, y_i, y))$. For any $j \in [1, N]$ and $s \in [1, T]$, we also define the weighted error $\epsilon_{s,j}$ as follows:

$$\epsilon_{s,j} = \frac{1}{2}\Big[1 - \underset{(i,y) \sim \mathcal{D}_s}{\mathrm{E}} \big[\mathsf{h}_j(x_i, y_i, y)\big]\Big]. \tag{12}$$

Figure 1 gives the pseudocode of the MDeepBoostSum algorithm. The details of the derivation of the expressions are given in Appendix E. In the special cases of the exponential loss ($\Phi(-u) = \exp(-u)$) or the logistic loss ($\Phi(-u) = \log_2(1 + \exp(-u))$), a closed-form expression is given for the step size (lines 6-8), which is the same in both cases (see Sections E.2.1 and E.2.2). In the generic case, the step size can be found using a line search or other numerical methods.

The algorithms presented above have several connections with other boosting algorithms, particularly in the absence of regularization. We discuss these connections in detail in Appendix K.

# 4 Experiments

The algorithms presented in the previous sections can be used with a variety of different base classifier sets. For our experiments, we used multi-class binary decision trees. A multi-class binary decision tree in dimension $d$ can be defined by a pair $(\mathsf{t}, \mathbf{h})$, where $\mathsf{t}$ is a binary tree with a variable-threshold question at each internal node, e.g., $X_j \le \theta$, $j \in [1, d]$, and $\mathbf{h} = (h_l)_{l \in \text{Leaves}(\mathsf{t})}$ a vector of distributions over the leaves $\text{Leaves}(\mathsf{t})$ of $\mathsf{t}$. At any leaf $l \in \text{Leaves}(\mathsf{t})$, $h_l(y) \in [0, 1]$ for all $y \in \mathcal{Y}$ and $\sum_{y \in \mathcal{Y}} h_l(y) = 1$. For convenience, we will denote by $\mathsf{t}(x)$ the leaf $l \in \text{Leaves}(\mathsf{t})$ associated to $x$ by $\mathsf{t}$. Thus, the score associated by $(\mathsf{t}, \mathbf{h})$ to a pair $(x, y) \in \mathcal{X} \times \mathcal{Y}$ is $h_l(y)$ where $l = \mathsf{t}(x)$.

Let $\mathcal{T}_n$ denote the family of all multi-class decision trees with $n$ internal nodes in dimension $d$. In Appendix J, we derive the following upper bound on the Rademacher complexity of $\mathcal{T}_n$:

$$\mathfrak{R}(\Pi_1(\mathcal{T}_n)) \le \sqrt{\frac{(4n+2)\log_2(d+2)\log(m+1)}{m}}. \tag{13}$$

All of the experiments in this section use $\mathcal{T}_n$ as the family of base hypothesis sets (parametrized by $n$). Since $\mathcal{T}_n$ is a very large hypothesis set when $n$ is large, for the sake of computational efficiency we make a few approximations. First, although our MDeepBoost algorithms were derived in terms of Rademacher complexity, we use the upper bound in Eq. (13) in place of the Rademacher complexity (thus, in Algorithm 1 we let $\Lambda_n = \lambda B_n + \beta$, where $B_n$ is the bound given in Eq. (13)). Secondly, instead of exhaustively searching for the best decision tree in $\mathcal{T}_n$ for each possible size $n$, we use the following greedy procedure: Given the best decision tree of size $n$ (starting with $n = 1$), we find the best decision tree of size $n+1$ that can be obtained by splitting one leaf, and continue this procedure until some maximum depth $K$. Decision trees are commonly learned in this manner, and so in this context our Rademacher-complexity-based bounds can be viewed as a novel stopping criterion for decision tree learning. Let $H_K^*$ be the set of trees found by the greedy algorithm just described. In each iteration $t$ of MDeepBoost, we select the best tree in the set $H_K^* \cup \{h_1, \ldots, h_{t-1}\}$, where $h_1, \ldots, h_{t-1}$ are the trees selected in previous iterations.

While we described many objective functions that can be used as the basis of a multi-class deep boosting algorithm, the experiments in this section focus on algorithms derived from $F_{\text{sum}}$. We also refer the reader to Table 3 in Appendix A for results of experiments with $F_{\text{compsum}}$ objective functions. The $F_{\text{sum}}$ and $F_{\text{compsum}}$ objectives combine several advantages that suggest they will perform well empirically. $F_{\text{sum}}$ is consistent and both $F_{\text{sum}}$ and $F_{\text{compsum}}$ are (by Theorem 4) $H$-consistent. Also, unlike $F_{\text{max}}$ both of these objectives are differentiable, and therefore the convergence guarantee in Theorem 2 applies. Our preliminary findings also indicate that algorithms based on $F_{\text{sum}}$ and $F_{\text{compsum}}$ objectives perform better than those derived from $F_{\text{max}}$ and $F_{\text{maxsum}}$. All of our objective functions require a choice for $\Phi$, the loss function. Since Cortes et al. [2014] reported comparable results for exponential and logistic loss for the binary version of DeepBoost, we let $\Phi$ be the exponential loss in all of our experiments with MDeepBoostSum. For MDeepBoostCompSum we select $\Phi_1(u) = \log_2(1 + u)$ and $\Phi_2(-u) = \exp(-u)$.

In our experiments, we used 8 UCI data sets: abalone, handwritten, letters, pageblocks, pendigits, satimage, statlog and yeast – see more details on these datasets in Table 4, Appendix L. In Appendix K, we explain that when $\lambda = \beta = 0$ then MDeepBoostSum is equivalent to AdaBoost.MR. Also, if we set $\lambda = 0$ and $\beta \ne 0$ then the resulting algorithm is an $L_1$-norm regularized variant of AdaBoost.MR. We compared MDeepBoostSum to these two algorithms, with the results also reported in Table 1 and Table 2 in Appendix A. Likewise, we compared MDeepBoostCompSum with multinomial (additive) logistic regression, LogReg, and its $L_1$-regularized version LogReg-L1, which, as discussed in Appendix K, are equivalent to MDeepBoostCompSum when $\lambda = \beta = 0$ and $\lambda = 0, \beta \ge 0$ respectively. Finally, we remark that it can be argued that the parameter optimization procedure (described below) significantly extends AdaBoost.MR since it effectively implements structural risk minimization: for each tree depth, the empirical error is minimized and we choose the depth to achieve the best generalization error.

All of these algorithms use maximum tree depth $K$ as a parameter. $L_1$-norm regularized versions admit two parameters: $K$ and $\beta \ge 0$. Deep boosting algorithms have a third parameter, $\lambda \ge 0$. To set these parameters, we used the following parameter optimization procedure: we randomly partitioned each dataset into 4 folds and, for each tuple $(\lambda, \beta, K)$ in the set of possible parameters (described below), we ran MDeepBoostSum, with a different assignment of folds to the training

Table 1: Empirical results for MDeepBoostSum, $\Phi = \exp$. AB stands for AdaBoost.

| abalone | AB.MR | AB.MR-L1 | MDeepBoost | handwritten | AB.MR | AB.MR-L1 | MDeepBoost |
|---|---|---|---|---|---|---|---|
| Error | 0.739 | 0.737 | 0.735 | Error | 0.024 | 0.025 | 0.021 |
| (std dev) | (0.0016) | (0.0065) | (0.0045) | (std dev) | (0.0011) | (0.0018) | (0.0015) |

| letters | AB.MR | AB.MR-L1 | MDeepBoost | pageblocks | AB.MR | AB.MR-L1 | MDeepBoost |
|---|---|---|---|---|---|---|---|
| Error | 0.065 | 0.059 | 0.058 | Error | 0.035 | 0.035 | 0.033 |
| (std dev) | (0.0018) | (0.0059) | (0.0039) | (std dev) | (0.0045) | (0.0031) | (0.0014) |

| pendigits | AB.MR | AB.MR-L1 | MDeepBoost | satimage | AB.MR | AB.MR-L1 | MDeepBoost |
|---|---|---|---|---|---|---|---|
| Error | 0.014 | 0.014 | 0.012 | Error | 0.112 | 0.117 | 0.117 |
| (std dev) | (0.0025) | (0.0013) | (0.0011) | (std dev) | (0.0123) | (0.0096) | (0.0087) |

| statlog | AB.MR | AB.MR-L1 | MDeepBoost | yeast | AB.MR | AB.MR-L1 | MDeepBoost |
|---|---|---|---|---|---|---|---|
| Error | 0.029 | 0.026 | 0.024 | Error | 0.415 | 0.410 | 0.407 |
| (std dev) | (0.0026) | (0.0071) | (0.0008) | (std dev) | (0.0353) | (0.0324) | (0.0282) |

set, validation set and test set for each run. Specifically, for each run $i \in \{0, 1, 2, 3\}$, fold $i$ was used for testing, fold $i + 1 \pmod 4$ was used for validation, and the remaining folds were used for training. For each run, we selected the parameters that had the lowest error on the validation set and then measured the error of those parameters on the test set. The average test error and the standard deviation of the test error over all 4 runs is reported in Table 1. Note that an alternative procedure to compare algorithms that is adopted in a number of previous studies of boosting [Li, 2009a,b, Sun et al., 2012] is to simply record the average test error of the best parameter tuples over all runs. While it is of course possible to overestimate the performance of a learning algorithm by optimizing hyperparameters on the test set, this concern is less valid when the size of the test set is large relative to the "complexity" of the hyperparameter space. We report results for this alternative procedure in Table 2 and Table 3, Appendix A.

For each dataset, the set of possible values for $\lambda$ and $\beta$ was initialized to $\{10^{-5}, 10^{-6}, \ldots, 10^{-10}\}$, and to $\{1, 2, 3, 4, 5\}$ for the maximum tree depth $K$. However, if we found an optimal parameter value to be at the end point of these ranges, we extended the interval in that direction (by an order of magnitude for $\lambda$ and $\beta$, and by 1 for the maximum tree depth $K$) and re-ran the experiments. We have also experimented with 200 and 500 iterations but we have observed that the errors do not change significantly and the ranking of the algorithms remains the same.

The results of our experiments show that, for each dataset, deep boosting algorithms outperform the other algorithms evaluated in our experiments. Let us point out that, even though not all of our results are statistically significant, MDeepBoostSum outperforms AdaBoost.MR and AdaBoost.MR-L1 (and, hence, effectively structural risk minimization) on each dataset. More importantly, for each dataset MDeepBoostSum outperforms other algorithms on most of the individual runs. Moreover, results for some datasets presented here (namely pendigits) appear to be state-of-the-art. We also refer our reader to experimental results summarized in Table 2 and Table 3 in Appendix A. These results provide further evidence in favor of DeepBoost algorithms. The consistent performance improvement by MDeepBoostSum over AdaBoost.MR or its L1-norm regularized variant shows the benefit of the new complexity-based regularization we introduced.

## 5   Conclusion

We presented new data-dependent learning guarantees for convex ensembles in the multi-class setting where the base classifier set is composed of increasingly complex sub-families, including very deep or complex ones. These learning bounds generalize to the multi-class setting the guarantees presented by Cortes et al. [2014] in the binary case. We also introduced and discussed several new multi-class ensemble algorithms benefiting from these guarantees and proved positive results for the $H$-consistency and convergence of several of them. Finally, we reported the results of several experiments with DeepBoost algorithms, and compared their performance with that of AdaBoost.MR and additive multinomial Logistic Regression and their $L_1$-regularized variants.

**Acknowledgments**

We thank Andres Muñoz Medina and Scott Yang for discussions and help with the experiments. This work was partly funded by the NSF award IIS-1117591 and supported by a NSERC PGS grant.

## Footnotes

[1] The condition $\sum_{t=1}^{T} \alpha_t = 1$ of Theorem 3 can be relaxed to $\sum_{t=1}^{T} \alpha_t \leq 1$. To see this, use for example a null hypothesis ($h_t = 0$ for some $t$).

[2]Note that this is a standard practice in the field of optimization. The optimization problem in (4) is equivalent to a vector optimization problem, where $(\sum_{i=1}^{m} 1_{\rho_f(x_i, y_i) \leq 1}, \sum_{t=1}^{T} \alpha_t r_t)$ is minimized over $\boldsymbol{\alpha}$. The latter problem can be scalarized leading to the introduction of a parameter $\lambda$ in (5).

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
