[Supplementary Material]

Table 2: Empirical results for MDeepBoostSum, $\Phi = \exp$. AB stands for AdaBoost.

| abalone | AB.MR | AB.MR-L1 | MDeepBoost |
|---|---|---|---|
| Error | 0.713 | 0.696 | **0.677** |
| (std dev) | (0.0130) | (0.0132) | (0.0092) |
| Avg tree size | 69.8 | 31.5 | 23.8 |
| Avg no. of trees | 17.9 | 13.3 | 15.3 |

| handwritten | AB.MR | AB.MR-L1 | MDeepBoost |
|---|---|---|---|
| Error | 0.016 | 0.011 | **0.009** |
| (std dev) | (0.0047) | (0.0026) | (0.0012) |
| Avg tree size | 187.3 | 240.6 | 203.0 |
| Avg no. of trees | 34.2 | 21.7 | 24.2 |

| letters | AB.MR | AB.MR-L1 | MDeepBoost |
|---|---|---|---|
| Error | 0.042 | 0.036 | **0.032** |
| (std dev) | (0.0023) | (0.0018) | (0.0016) |
| Avg tree size | 1942.6 | 1903.8 | 1914.6 |
| Avg no. of trees | 24.2 | 24.4 | 23.3 |

| pageblocks | AB.MR | AB.MR-L1 | MDeepBoost |
|---|---|---|---|
| Error | 0.020 | 0.017 | **0.013** |
| (std dev) | (0.0037) | (0.0021) | (0.0027) |
| Avg tree size | 134.8 | 118.3 | 124.9 |
| Avg no. of trees | 8.5 | 14.3 | 6.6 |

| pendigits | AB.MR | AB.MR-L1 | MDeepBoost |
|---|---|---|---|
| Error | 0.008 | 0.006 | **0.004** |
| (std dev) | (0.0015) | (0.0023) | (0.0011) |
| Avg tree size | 272.5 | 283.3 | 259.2 |
| Avg no. of trees | 23.2 | 19.8 | 21.4 |

| satimage | AB.MR | AB.MR-L1 | MDeepBoost |
|---|---|---|---|
| Error | 0.089 | 0.081 | **0.073** |
| (std dev) | (0.0062) | (0.0040) | (0.0045) |
| Avg tree size | 557.9 | 478.8 | 535.6 |
| Avg no. of trees | 7.6 | 7.3 | 7.6 |

| statlog | AB.MR | AB.MR-L1 | MDeepBoost |
|---|---|---|---|
| Error | 0.011 | 0.006 | **0.004** |
| (std dev) | (0.0059) | (0.0035) | (0.0030) |
| Avg tree size | 74.8 | 79.2 | 61.8 |
| Avg no. of trees | 23.2 | 17.5 | 17.6 |

| yeast | AB.MR | AB.MR-L1 | MDeepBoost |
|---|---|---|---|
| Error | 0.388 | 0.376 | **0.352** |
| (std dev) | (0.0392) | (0.0431) | (0.0402) |
| Avg tree size | 100.6 | 111.7 | 71.4 |
| Avg no. of trees | 8.7 | 6.5 | 7.7 |

# A    Additional Experiments

In this section, we present some further experimental results for MDeepBoostSum and MDeep-BoostCompSum algorithms. Recall that the results of Table 1 were obtained using the following parameter optimization procedure. We randomly partitioned each dataset into 4 folds and, for each tuple $(\lambda, \beta, K)$ in the set of possible parameters (described below), we ran MDeepBoostSum, with a different assignment of folds to the training set, validation set and test set for each run. Specifically, for each run $i \in \{0, 1, 2, 3\}$, fold $i$ was used for testing, fold $i + 1 \pmod 4$ was used for validation, and the remaining folds were used for training. For each run, we selected the parameters that had the lowest error on the validation set and then measured the error of those parameters on the test set. The average error and the standard deviation of the error over all 4 runs is reported in Table 1. We noted that there is an alternative procedure to compare algorithms that is adopted in a number of previous studies of boosting [Li, 2009a,b, Sun et al., 2012] which is to simply record the average test error of the best parameter tuples over all runs. We argued that as the size of the validation set grows, the errors obtained via this procedure should converge to the true generalization error of the algorithm. The results for this alternative procedure are shown in Table 2 and Table 3.

Observe that once again the results of our experiments show that for each dataset deep boosting algorithms outperform their shallow rivals. Moreover, all of our results are statistically significant, at 5% level using one-sided, paired $t$-test. This provides further empirical evidence in favor of DeepBoost algorithms.

# B    Proof of Theorem 1

Our proof makes use of existing methods for deriving Rademacher complexity bounds [Koltchinskii and Panchenko, 2002] and a proof technique used in [Schapire et al., 1997].

**Theorem 1.** *Assume $p > 1$ and let $H_1, \ldots, H_p$ be $p$ families of functions mapping from $\mathcal{X} \times \mathcal{Y}$ to $[0, 1]$. Fix $\rho > 0$. Then, for any $\delta > 0$, with probability at least $1 - \delta$ over the choice of a sample $S$ of size $m$ drawn i.i.d. according to $\mathcal{D}$, the following inequality holds for all $f = \sum_{t=1}^{T} \alpha_t h_t \in \mathcal{F}$:*

$$R(f) \leq \widehat{R}_{S,\rho}(f) + \frac{8c}{\rho} \sum_{t=1}^{T} \alpha_t \mathfrak{R}_m(\Pi_1(H_{k_t})) + \frac{2}{c\rho} \sqrt{\frac{\log p}{m}} + \sqrt{\left\lceil \frac{4}{\rho^2} \log\left(\frac{c^2 \rho^2 m}{4 \log p}\right)\right\rceil \frac{\log p}{m} + \frac{\log \frac{2}{\delta}}{2m}},$$

*Thus,* $R(f) \leq \widehat{R}_{S,\rho}(f) + \frac{8c}{\rho} \sum_{t=1}^{T} \alpha_t \mathfrak{R}_m(H_{k_t}) + O\left(\sqrt{\frac{\log p}{\rho^2 m} \log\left[\frac{\rho^2 c^2 m}{4 \log p}\right]}\right).$

Table 3: Empirical results for MDeepBoostCompSum, $\Phi_1(u) = \log_2(1+u)$ and $\Phi_2 = \exp(u+1)$.

| abalone | LogReg | LogReg-L1 | MDeepBoost |
|---|---|---|---|
| Error | 0.710 | 0.700 | **0.687** |
| (std dev) | (0.0170) | (0.0102) | (0.0104) |
| Avg tree size | 162.1 | 156.5 | 28.0 |
| Avg no. of trees | 22.2 | 9.8 | 10.2 |

| handwritten | LogReg | LogReg-L1 | MDeepBoost |
|---|---|---|---|
| Error | 0.016 | 0.012 | **0.008** |
| (std dev) | (0.0031) | (0.0020) | (0.0024) |
| Avg tree size | 237.7 | 186.5 | 153.8 |
| Avg no. of trees | 32.3 | 32.8 | 35.9 |

| letters | LogReg | LogReg-L1 | MDeepBoost |
|---|---|---|---|
| Error | 0.043 | 0.038 | **0.035** |
| (std dev) | (0.0018) | (0.0012) | (0.0012) |
| Avg tree size | 1986.5 | 1759.5 | 1807.3 |
| Avg no. of trees | 25.5 | 29.0 | 27.2 |

| pageblocks | LogReg | LogReg-L1 | MDeepBoost |
|---|---|---|---|
| Error | 0.019 | 0.016 | **0.012** |
| (std dev) | (0.0035) | (0.0025) | (0.0022) |
| Avg tree size | 127.4 | 151.7 | 147.9 |
| Avg no. of trees | 4.5 | 6.8 | 7.4 |

| pendigits | LogReg | LogReg-L1 | MDeepBoost |
|---|---|---|---|
| Error | 0.009 | 0.007 | **0.005** |
| (std dev) | (0.0021) | (0.0014) | (0.0012) |
| Avg tree size | 306.3 | 277.1 | 262.7 |
| Avg no. of trees | 21.9 | 20.8 | 19.7 |

| satimage | LogReg | LogReg-L1 | MDeepBoost |
|---|---|---|---|
| Error | 0.091 | 0.082 | **0.074** |
| (std dev) | (0.0066) | (0.0057) | (0.0056) |
| Avg tree size | 412.6 | 454.6 | 439.6 |
| Avg no. of trees | 6.0 | 5.8 | 5.8 |

| statlog | LogReg | LogReg-L1 | MDeepBoost |
|---|---|---|---|
| Error | 0.012 | 0.006 | **0.002** |
| (std dev) | (0.0054) | (0.0020) | (0.0022) |
| Avg tree size | 74.3 | 71.6 | 65.4 |
| Avg no. of trees | 22.3 | 20.6 | 17.5 |

| yeast | LogReg | LogReg-L1 | MDeepBoost |
|---|---|---|---|
| Error | 0.381 | 0.375 | **0.354** |
| (std dev) | (0.0467) | (0.0458) | (0.0468) |
| Avg tree size | 103.9 | 83.3 | 117.2 |
| Avg no. of trees | 14.1 | 9.3 | 9.3 |

*Proof.* For a fixed $\mathbf{h} = (h_1, \ldots, h_T)$, any $\boldsymbol{\alpha}$ in the probability simplex $\Delta$ defines a distribution over $\{h_1, \ldots, h_T\}$. Sampling from $\{h_1, \ldots, h_T\}$ according to $\boldsymbol{\alpha}$ and averaging leads to functions $g$ of the form $g = \frac{1}{n}\sum_{i=1}^{T} n_t h_t$ for some $\mathbf{n} = (n_1, \ldots, n_T)$, with $\sum_{t=1}^{T} n_t = n$, and $h_t \in H_{k_t}$.

For any $\mathbf{N} = (N_1, \ldots, N_p)$ with $|\mathbf{N}| = n$, we consider the family of functions

$$G_{\mathcal{F},\mathbf{N}} = \left\{ \frac{1}{n}\sum_{k=1}^{p}\sum_{j=1}^{N_k} h_{k,j} \mid \forall (k,j) \in [p] \times [N_k], h_{k,j} \in H_k \right\},$$

and the union of all such families $G_{\mathcal{F},n} = \bigcup_{|\mathbf{N}|=n} G_{\mathcal{F},\mathbf{N}}$. Fix $\rho > 0$. For a fixed $\mathbf{N}$, the Rademacher complexity of $\Pi_1(G_{\mathcal{F},\mathbf{N}})$ can be bounded as follows for any $m \geq 1$: $\mathfrak{R}_m(\Pi_1(G_{\mathcal{F},\mathbf{N}})) \leq \frac{1}{n}\sum_{k=1}^{p} N_k \mathfrak{R}_m(\Pi_1(H_k))$. Thus, by Theorem 3, the following multi-class margin-based Rademacher complexity bound holds. For any $\delta > 0$, with probability at least $1 - \delta$, for all $g \in G_{\mathcal{F},\mathbf{N}}$,

$$R_\rho(g) - \widehat{R}_{S,\rho}(g) \leq \frac{1}{n}\frac{4c}{\rho}\sum_{k=1}^{p} N_k \mathfrak{R}_m(\Pi_1(H_k)) + \sqrt{\frac{\log\frac{1}{\delta}}{2m}}.$$

Since there are at most $p^n$ possible $p$-tuples $\mathbf{N}$ with $|\mathbf{N}| = n$,[3] by the union bound, for any $\delta > 0$, with probability at least $1 - \delta$, for all $g \in G_{\mathcal{F},n}$, we can write

$$R_\rho(g) - \widehat{R}_{S,\rho}(g) \leq \frac{1}{n}\frac{4c}{\rho}\sum_{k=1}^{p} N_k \mathfrak{R}_m(\Pi_1(H_k)) + \sqrt{\frac{\log\frac{p^n}{\delta}}{2m}}.$$

Thus, with probability at least $1-\delta$, for all functions $g = \frac{1}{n}\sum_{i=1}^{T} n_t h_t$ with $h_t \in H_{k_t}$, the following inequality holds

$$R_\rho(g) - \widehat{R}_{S,\rho}(g) \leq \frac{1}{n}\frac{4c}{\rho}\sum_{k=1}^{p}\sum_{t:k_t=k} n_t \mathfrak{R}_m(\Pi_1(H_{k_t})) + \sqrt{\frac{\log\frac{p^n}{\delta}}{2m}}.$$

Taking the expectation with respect to $\boldsymbol{\alpha}$ and using $\mathrm{E}_{\boldsymbol{\alpha}}[n_t/n] = \alpha_t$, we obtain that for any $\delta > 0$, with probability at least $1 - \delta$, for all $g$, we can write

$$\mathop{\mathrm{E}}_{\boldsymbol{\alpha}}[R_\rho(g) - \widehat{R}_{S,\rho}(g)] \leq \frac{4c}{\rho}\sum_{t=1}^{T} \alpha_t \mathfrak{R}_m(\Pi_1(H_{k_t})) + \sqrt{\frac{\log\frac{p^n}{\delta}}{2m}}.$$

Fix $n \geq 1$. Then, for any $\delta_n > 0$, with probability at least $1 - \delta_n$,

$$\mathop{\mathrm{E}}_{\boldsymbol{\alpha}}[R_{\rho/2}(g) - \widehat{R}_{S,\rho/2}(g)] \leq \frac{8c}{\rho} \sum_{t=1}^{T} \alpha_t \mathfrak{R}_m(\Pi_1(H_{k_t})) + \sqrt{\frac{\log \frac{p^n}{\delta_n}}{2m}}.$$

Choose $\delta_n = \frac{\delta}{2p^{n-1}}$ for some $\delta > 0$, then for $p \geq 2$, $\sum_{n \geq 1} \delta_n = \frac{\delta}{2(1-1/p)} \leq \delta$. Thus, for any $\delta > 0$ and any $n \geq 1$, with probability at least $1 - \delta$, the following holds for all $g$:

$$\mathop{\mathrm{E}}_{\boldsymbol{\alpha}}[R_{\rho/2}(g) - \widehat{R}_{S,\rho/2}(g)] \leq \frac{8c}{\rho} \sum_{t=1}^{T} \alpha_t \mathfrak{R}_m(\Pi_1(H_{k_t})) + \sqrt{\frac{\log \frac{2p^{2n-1}}{\delta}}{2m}}. \tag{14}$$

Now, for any $f = \sum_{t=1}^{T} \alpha_t h_t \in \mathcal{F}$ and any $g = \frac{1}{n} \sum_{i=1}^{T} n_t h_t$, we can upper-bound $R(f) = \Pr_{(x,y) \sim \mathcal{D}}[\rho_f(x,y) \leq 0]$, the generalization error of $f$, as follows:

$$\begin{aligned}
R(f) &= \mathop{\mathrm{Pr}}_{(x,y) \sim \mathcal{D}}[\rho_f(x,y) - \rho_g(x,y) + \rho_g(x,y) \leq 0] \\
&\leq \Pr[\rho_f(x,y) - \rho_g(x,y) < -\rho/2] + \Pr[\rho_g(x,y) \leq \rho/2] \\
&= \Pr[\rho_f(x,y) - \rho_g(x,y) < -\rho/2] + R_{\rho/2}(g).
\end{aligned}$$

We can also write

$$\widehat{R}_{\rho/2}(g) = \widehat{R}_{S,\rho/2}(g - f + f) \leq \mathop{\mathrm{Pr}}_{(x,y) \sim S}[\rho_g(x,y) - \rho_f(x,y) < -\rho/2] + \widehat{R}_{S,\rho}(f).$$

Combining these inequalities yields

$$\begin{aligned}
\mathop{\mathrm{Pr}}_{(x,y) \sim \mathcal{D}}[\rho_f(x,y) \leq 0] - \widehat{R}_{S,\rho}(f) &\leq \mathop{\mathrm{Pr}}_{(x,y) \sim \mathcal{D}}[\rho_f(x,y) - \rho_g(x,y) < -\rho/2] \\
&\quad + \mathop{\mathrm{Pr}}_{(x,y) \sim S}[\rho_g(x,y) - \rho_f(x,y) < -\rho/2] + R_{\rho/2}(g) - \widehat{R}_{S,\rho/2}(g).
\end{aligned}$$

Taking the expectation with respect to $\boldsymbol{\alpha}$ yields

$$\begin{aligned}
R(f) - \widehat{R}_{S,\rho}(f) &\leq \mathop{\mathrm{E}}_{(x,y) \sim \mathcal{D}, \boldsymbol{\alpha}}[\mathbf{1}_{\rho_f(x,y) - \rho_g(x,y) < -\rho/2}] \\
&\quad + \mathop{\mathrm{E}}_{(x,y) \sim S, \boldsymbol{\alpha}}[\mathbf{1}_{\rho_g(x,y) - \rho_f(x,y) < -\rho/2}] + \mathop{\mathrm{E}}_{\boldsymbol{\alpha}}[R_{\rho/2}(g) - \widehat{R}_{S,\rho/2}(g)]. \tag{15}
\end{aligned}$$

Fix $(x,y)$ and for any function $\varphi \colon \mathcal{X} \times \mathcal{Y} \to [0,1]$ define $y'_\varphi$ as follows: $y'_\varphi = \mathrm{argmax}_{y' \neq y} \varphi(x,y)$. For any $g$, by definition of $\rho_g$, we can write $\rho_g(x,y) \leq g(x,y) - g(x,y'_f)$. In light of this inequality and Hoeffding's bound, the following holds:

$$\begin{aligned}
\mathop{\mathrm{E}}_{\boldsymbol{\alpha}}[\mathbf{1}_{\rho_f(x,y) - \rho_g(x,y) < -\rho/2}] &= \mathop{\mathrm{Pr}}_{\boldsymbol{\alpha}}[\rho_f(x,y) - \rho_g(x,y) < -\rho/2] \\
&\leq \mathop{\mathrm{Pr}}_{\boldsymbol{\alpha}}\left[\big(f(x,y) - f(x,y'_f)\big) - \big(g(x,y) - g(x,y'_f)\big) < -\rho/2\right] \\
&\leq e^{-n\rho^2/8}.
\end{aligned}$$

Similarly, for any $g$, we can write $\rho_f(x,y) \leq f(x,y) - f(x,y'_g)$. Using this inequality, the union bound and Hoeffding's bound, the other expectation term appearing on the right-hand side of (15) can be bounded as follows:

$$\begin{aligned}
\mathop{\mathrm{E}}_{\boldsymbol{\alpha}}[\mathbf{1}_{\rho_g(x,y) - \rho_f(x,y) < -\rho/2}] &= \mathop{\mathrm{Pr}}_{\boldsymbol{\alpha}}[\rho_g(x,y) - \rho_f(x,y) < -\rho/2] \\
&\leq \mathop{\mathrm{Pr}}_{\boldsymbol{\alpha}}\left[\big(g(x,y) - g(x,y'_g)\big) - \big(f(x,y) - f(x,y'_g)\big) < -\rho/2\right] \\
&\leq \sum_{y' \neq y} \mathop{\mathrm{Pr}}_{\boldsymbol{\alpha}}\left[\big(g(x,y) - g(x,y')\big) - \big(f(x,y) - f(x,y')\big) < -\rho/2\right] \\
&\leq (c-1)e^{-n\rho^2/8}.
\end{aligned}$$

Thus, for any fixed $f \in \mathcal{F}$, we can write

$$R(f) - \widehat{R}_{S,\rho}(f) \leq ce^{-n\rho^2/8} + \mathop{\mathrm{E}}_{\boldsymbol{\alpha}}[R_{\rho/2}(g) - \widehat{R}_{S,\rho/2}(g)].$$

Therefore, the following inequality holds:

$$\sup_{f \in \mathcal{F}} R(f) - \widehat{R}_{S,\rho}(f) \leq ce^{-n\rho^2/8} + \sup_g \mathrm{E}_{\boldsymbol{\alpha}}[R_{\rho/2}(g) - \widehat{R}_{S,\rho/2}(g)],$$

and, in view of (14), for any $\delta > 0$ and any $n \geq 1$, with probability at least $1 - \delta$, the following holds for all $f \in \mathcal{F}$:

$$R(f) - \widehat{R}_{S,\rho}(f) \leq \frac{8c}{\rho} \sum_{t=1}^{T} \alpha_t \mathfrak{R}_m(\Pi_1(H_{k_t})) + ce^{-\frac{n\rho^2}{8}} + \sqrt{\frac{(2n-1)\log p + \log \frac{2}{\delta}}{2m}}.$$

Choosing $n = \left\lceil \frac{4}{\rho^2} \log \left( \frac{c^2 \rho^2 m}{4 \log p} \right) \right\rceil$ yields the following inequality:[4]

$$R(f) - \widehat{R}_{S,\rho}(f) \leq \frac{8c}{\rho} \sum_{t=1}^{T} \alpha_t \mathfrak{R}_m(\Pi_1(H_{k_t})) + \frac{2}{c\rho} \sqrt{\frac{\log p}{m}} + \sqrt{\left\lceil \frac{4}{\rho^2} \log \left( \frac{c^2 \rho^2 m}{4 \log p} \right) \right\rceil \frac{\log p}{m} + \frac{\log \frac{2}{\delta}}{2m}},$$

and concludes the proof. $\qquad\square$

Our proof of Theorem 1 made use of the following general multi-class learning bound, which admits a more favorable dependency on the number of classes $c$ than the existing multi-class Rademacher complexity bounds of [Koltchinskii and Panchenko, 2002, Mohri et al., 2012].

**Theorem 3.** *Let $G$ be a family of hypotheses mapping $\mathcal{X} \times \mathcal{Y}$ to $\mathbb{R}$, with $\mathcal{Y} = \{1, \ldots, c\}$. Fix $\rho > 0$. Then, for any $\delta > 0$, with probability at least $1 - \delta > 0$, the following bound holds for all $g \in G$:*

$$R(g) \leq \widehat{R}_{S,\rho}(g) + \frac{4c}{\rho} \mathfrak{R}_m(\Pi_1(G)) + \sqrt{\frac{\log \frac{1}{\delta}}{2m}},$$

*where $\Pi_1(G) = \{(x, y) \mapsto g(x, y) : y \in \mathcal{Y}, g \in G\}$.*

*Proof.* We will need the following definition for this proof:

$$\rho_g(x, y) = \min_{y' \neq y}(g(x, y) - g(x, y'))$$
$$\rho_{\theta, g}(x, y) = \min_{y'}(g(x, y) - g(x, y') + \theta 1_{y'=y}),$$

where $\theta > 0$ is an arbitrary constant. Observe that $\mathrm{E}[1_{\rho_g(x,y) \leq 0}] \leq \mathrm{E}[1_{\rho_{\theta,g}(x,y) \leq 0}]$ since the inequality $\rho_{\theta,g}(x, y) \leq \rho_g(x, y)$ holds for all $(x, y) \in \mathcal{X} \times \mathcal{Y}$:

$$\rho_{\theta,g}(x, y) = \min_{y'} \left( g(x, y) - g(x, y') + \theta 1_{y'=y} \right)$$
$$\leq \min_{y' \neq y} \left( g(x, y) - g(x, y') + \theta 1_{y'=y} \right)$$
$$= \min_{y' \neq y} \left( g(x, y) - g(x, y') \right) = \rho_g(x, y),$$

where the inequality follows from taking the minimum over a smaller set.

Let $\Phi_\rho$ be the margin loss function defined for all $u \in \mathbb{R}$ by $\Phi_\rho(u) = 1_{u \leq 0} + (1 - \frac{u}{\rho})1_{0 < u \leq \rho}$. We also let $\widetilde{G} = \{(x, y) \mapsto \rho_{\theta,g}(x, y) : g \in G\}$ and $\widetilde{\mathcal{G}} = \{\Phi_\rho \circ \widetilde{g} : \widetilde{g} \in \widetilde{G}\}$. By the standard Rademacher complexity bound [Koltchinskii and Panchenko, 2002, Mohri et al., 2012], for any $\delta > 0$, with probability at least $1 - \delta$, the following holds for all $g \in G$:

$$R(g) \leq \frac{1}{m} \sum_{i=1}^{m} \Phi_\rho(\rho_{\theta,g}(x_i, y_i)) + 2\mathfrak{R}_m(\widetilde{\mathcal{G}}) + \sqrt{\frac{\log \frac{1}{\delta}}{2m}}.$$

Fixing $\theta = 2\rho$, we observe that $\Phi_\rho(\rho_{\theta,g}(x_i, y_i)) = \Phi_\rho(\rho_g(x_i, y_i)) \leq 1_{\rho_g(x_i,y_i) \leq \rho}$. Indeed, either $\rho_{\theta,g}(x_i, y_i) = \rho_g(x_i, y_i)$ or $\rho_{\theta,g}(x_i, y_i) = 2\rho \leq \rho_g(x_i, y_i)$, which implies the desired result. Talagrand's lemma [Ledoux and Talagrand, 1991, Mohri et al., 2012] yields $\mathfrak{R}_m(\widetilde{\mathcal{G}}) \leq \frac{1}{\rho}\mathfrak{R}_m(\widetilde{G})$ since $\Phi_\rho$ is a $\frac{1}{\rho}$-Lipschitz function. Therefore, for any $\delta > 0$, with probability at least $1 - \delta$, for all $g \in G$:

$$R(g) \leq R_{S,\rho}(g) + \frac{2}{\rho}\mathfrak{R}_m(\widetilde{G}) + \sqrt{\frac{\log\frac{1}{\delta}}{2m}}.$$

and to complete the proof it suffices to show that $\mathfrak{R}_m(\widetilde{G}) \leq 2c\mathfrak{R}_m(\Pi_1(G))$.

Here $\mathfrak{R}_m(\widetilde{G})$ can be upper-bounded as follows:

$$\mathfrak{R}_m(\widetilde{G}) = \frac{1}{m} \mathop{\mathrm{E}}_{S,\sigma}\left[\sup_{g \in G} \sum_{i=1}^m \sigma_i(g(x_i, y_i) - \max_y(g(x_i, y) - 2\rho 1_{y=y_i}))\right]$$

$$\leq \frac{1}{m} \mathop{\mathrm{E}}_{S,\sigma}\left[\sup_{g \in G} \sum_{i=1}^m \sigma_i g(x_i, y_i)\right] + \frac{1}{m} \mathop{\mathrm{E}}_{S,\sigma}\left[\sup_{g \in G} \sum_{i=1}^m \sigma_i \max_y(g(x_i, y) - 2\rho 1_{y=y_i})\right].$$

Now we bound the second term above. Observe that

$$\frac{1}{m} \mathop{\mathrm{E}}_\sigma\left[\sup_{g \in G} \sum_{i=1}^m \sigma_i g(x_i, y_i)\right] = \frac{1}{m} \mathop{\mathrm{E}}_\sigma\left[\sup_{g \in G} \sum_{i=1}^m \sum_{y \in \mathcal{Y}} \sigma_i g(x_i, y) 1_{y_i = y}\right]$$

$$\leq \frac{1}{m} \sum_{y \in \mathcal{Y}} \mathop{\mathrm{E}}_\sigma\left[\sup_{g \in G} \sum_{i=1}^m \sigma_i g(x_i, y) 1_{y_i = y}\right]$$

$$= \sum_{y \in \mathcal{Y}} \frac{1}{m} \mathop{\mathrm{E}}_\sigma\left[\sup_{g \in G} \sum_{i=1}^m \sigma_i g(x_i, y)\left(\frac{\epsilon_i}{2} + \frac{1}{2}\right)\right],$$

where $\epsilon_i = 2 \cdot 1_{y_i = y} - 1$. Since $\epsilon_i \in \{-1, +1\}$, $\sigma_i$ and $\sigma_i \epsilon_i$ admit the same distribution and, for any $y \in \mathcal{Y}$, each of the terms of the right-hand side can be bounded as follows:

$$\frac{1}{m} \mathop{\mathrm{E}}_\sigma\left[\sup_{g \in G} \sum_{i=1}^m \sigma_i g(x_i, y)\left(\frac{\epsilon_i}{2} + \frac{1}{2}\right)\right]$$

$$\leq \frac{1}{2m} \mathop{\mathrm{E}}_\sigma\left[\sup_{g \in G} \sum_{i=1}^m \sigma_i \epsilon_i g(x_i, y)\right] + \frac{1}{2m} \mathop{\mathrm{E}}_\sigma\left[\sup_{g \in G} \sum_{i=1}^m \sigma_i g(x_i, y)\right]$$

$$\leq \widehat{\mathfrak{R}}_m(\Pi_1(G)).$$

Thus, we can write $\frac{1}{m} \mathop{\mathrm{E}}_{S,\sigma}\left[\sup_{g \in G} \sum_{i=1}^m \sigma_i g(x_i, y_i)\right] \leq c\mathfrak{R}_m(\Pi_1(G))$. To bound the second term, we first apply Lemma 8.1 of Mohri et al. [2012] that immediately yields that

$$\frac{1}{m} \mathop{\mathrm{E}}_{S,\sigma}\left[\sup_{g \in G} \sum_{i=1}^m \sigma_i \max_y(g(x_i, y) - 2\rho 1_{y=y_i})\right] \leq \sum_{y \in \mathcal{Y}} \frac{1}{m} \mathop{\mathrm{E}}_{S,\sigma}\left[\sup_{g \in G} \sum_{i=1}^m \sigma_i(g(x_i, y) - 2\rho 1_{y=y_i})\right]$$

and since Rademacher variables are mean zero, we observe that

$$\mathop{\mathrm{E}}_{S,\sigma}\left[\sup_{g \in G} \sum_{i=1}^m \sigma_i(g(x_i, y) - 2\rho 1_{y=y_i})\right] = \mathop{\mathrm{E}}_{S,\sigma}\left[\sup_{g \in G}\left(\sum_{i=1}^m \sigma_i g(x_i, y)\right) - 2\rho \sum_{i=1}^m \sigma_i 1_{y=y_i}\right]$$

$$= \mathop{\mathrm{E}}_{S,\sigma}\left[\sup_{g \in G} \sum_{i=1}^m \sigma_i g(x_i, y)\right] \leq \mathfrak{R}_m(\Pi_1(G))$$

which completes the proof. $\qquad\square$

# C    Consistency

In this section, we discuss several questions related to consistency. In the multi-class setting where $c$ scoring functions are used, a loss function $L$ is defined over $\mathcal{Y} \times \mathcal{Y}^c$ which associates to a class $y \in \mathcal{Y}$ and class scores $s_1, \ldots, s_c$ the real number $L(y, s_1, \ldots, s_c)$. Consistency has often served as a guide for the selection of a loss function. Informally, a loss function is consistent if minimizing it results in a classifier whose accuracy is close to that of the Bayes classifier. Tewari and Bartlett [2007] (see also [Zhang, 2004a,b]) showed that, in general, loss functions of the form $(s_1, \ldots, s_c, y) \mapsto \Phi(-(s_y - \max_{y' \neq y} s_{y'}))$ are not consistent, while those that can be written as $(s_1, \ldots, s_c, y) \mapsto \sum_{y' \neq y} \Phi(-(s_y - s_{y'}))$ are consistent under some additional regularity assumptions on $\Phi$. This may suggest that solving the optimization problem associated to $F_{\mathrm{sum}}$ may result in a more accurate classifier than the solution of $\min_{\boldsymbol{\alpha} \geq 0} F_{\mathrm{max}}(\boldsymbol{\alpha})$.[5]

However, the notion of loss consistency does not take into account the hypothesis set $H$ used since it assumes an optimization carried out over the set of all measurable functions. Long and Servedio [2013] proposed instead a notion of $H$-consistency precisely meant to take the hypothesis set used into consideration. They showed empirically that using loss functions that are $H$-consistent can lead to significantly better performances than using a loss function known to be consistent. Informally, a loss function is said to be $H$-consistent if minimizing it over $H$ results in a classifier achieving a generalization error close to that of the best classifier in $H$.

More formally, $L$ is said to be *realizable $H$-consistent* [Long and Servedio, 2013] if for any distribution $\mathcal{D}$ over $\mathcal{X} \times \mathcal{Y}$ realizable with respect to $H$ and any $\epsilon > 0$, there exists $\delta > 0$ such that if $|\operatorname{E}_{(x,y)\sim\mathcal{D}}[L(y, h(x,1), \ldots, h(x,c))] - \inf_{h \in H} \operatorname{E}_{(x,y)\sim\mathcal{D}}[L(y, h(x,1), \ldots, h(x,c))]| \leq \delta$, then $\operatorname{E}_{(x,y)\sim\mathcal{D}}[1_{\rho_h(x,y)\leq 0}] \leq \epsilon$. The following is an extension of a result of Long and Servedio [2013] to our setting.

**Theorem 4.** *Let $u \mapsto \Phi(-u)$ be a non-increasing function upper-bounding $u \mapsto 1_{u \leq 0}$, bounded over $\mathbb{R}_+$, and such that $\lim_{u \to \infty} \Phi(-u) = 0$, and let $H$ be a family of functions mapping $\mathcal{X} \times \mathcal{Y}$ to $\mathbb{R}$ closed under multiplication by a positive scalar ($H$ is a cone). Then, the loss functions $(s_1, \ldots, s_c, y) \mapsto \Phi(-(s_y - \max_{y' \neq y} s_{y'}))$ and $(s_1, \ldots, s_c, y) \mapsto \sum_{y' \neq y} \Phi(-(s_y - s_{y'}))$ are realizable $H$-consistent. Moreover, if $\Phi = \Phi_1 \circ \Phi_2$ with non-decreasing $\Phi_1$ and $\Phi_2$ verifying $\lim_{u \to 0} \Phi_1(u) = \lim_{u \to \infty} \Phi_2(-u) = 0$ then the loss function $(s_1, \ldots, s_c, y) \mapsto \Phi_1\left(\sum_{y' \neq y} \Phi_2(-(s_y - s_{y'}))\right)$ is also realizable $H$-consistent.*

*Proof.* Let $\mathcal{D}$ be a distribution for which $h^* \in H$ achieves zero error, thus $\rho_{h^*}(x,y) > 0$ for all $(x,y)$ in the support of $\mathcal{D}$. Fix $\epsilon > 0$ and assume that $|\operatorname{E}_{(x,y)\sim\mathcal{D}}[\Phi(-\rho_h(x,y))] - \inf_{h \in H} \operatorname{E}_{(x,y)\sim\mathcal{D}}[\Phi(-\rho_h(x,y))]| \leq \epsilon$ for some $h \in H$. Then, since $1_{u \leq 0} \leq \Phi(-u)$ and since $\eta h^*$ is in $H$ for any $\eta > 0$, the following holds for any $\eta > 0$:

$$
\begin{aligned}
\operatorname*{E}_{(x,y)\sim\mathcal{D}}[1_{\rho_h(x,y)\leq 0}] &\leq \operatorname*{E}_{(x,y)\sim\mathcal{D}}[\Phi(-\rho_h(x,y))] \\
&\leq \operatorname*{E}_{(x,y)\sim\mathcal{D}}[\Phi(-\rho_{\eta h^*}(x,y))] + \epsilon = \operatorname*{E}_{(x,y)\sim\mathcal{D}}[\Phi(-\eta\rho_{h^*}(x,y))] + \epsilon.
\end{aligned}
$$

Since $\Phi(-u)$ is bounded for $u \geq 0$, by the Lebesgue dominated convergence theorem, $\lim_{\eta \to \infty} \operatorname{E}_{(x,y)\sim\mathcal{D}}[\Phi(-\rho_{\eta h^*}(x,y))] = 0$, which proves the first statement of the theorem. Now suppose that

$$
\left| \operatorname*{E}_{(x,y)\sim\mathcal{D}}\left[ \sum_{y' \neq y} \Phi((-(h(x,y) - h(x,y')))) \right] - \inf_{h \in H} \operatorname*{E}_{(x,y)\sim\mathcal{D}}\left[ \sum_{y' \neq y} \Phi((-(h(x,y) - h(x,y')))) \right] \right| \leq \epsilon
$$

for some $h \in H$. Using $1_{u \leq 0} \leq \Phi(-u)$, upper-bounding the maximum by a sum, and using the fact that $\eta h^*$ is in $H$ for any $\eta > 0$, the following holds for any $\eta > 0$:

$$
\operatorname{E}[1_{\rho_h(x,y)\leq 0}] \leq \operatorname{E}\left[ \sum_{y' \neq y} \Phi(-(h(x,y) - h(x,y'))) \right] \leq \operatorname{E}\left[ \sum_{y' \neq y} \Phi(-\eta(h^*(x,y) - h^*(x,y'))) \right] + \epsilon.
$$

---

Since by assumption $\rho_{h^*}(x,y) > 0$, the inequality $(h^*(x,y) - h^*(x,y')) > 0$ holds for all $y' \neq y$ by assumption. Therefore, applying the Lebesgue dominated convergence theorem as before yields the second statement of the theorem. The last statement can be proven in a similar way. □

The conditions of the theorem hold in particular for the exponential and the logistic functions and $H = \text{conv}(\bigcup_{k=1}^{p} H_k)$. Thus, the theorem shows that the loss functions associated to $F_{\max}$ and $F_{\text{sum}}$ are realizable $H$-consistent in the common cases where the exponential or logistic losses are used. Similarly, it shows that in the common case where $\Phi_1(u) = \log(1 + u)$ and $\Phi_2(u) = \exp(u + 1)$, the loss function associated to $F_{\text{compsum}}$ is $H$-consistent.

## D  Relationships between objective functions

One caveat of $F_{\max}$ is that it is not differentiable. In some cases, it may be desirable to deal with a somewhat simpler optimization problem with a differentiable objective function. As it was already observed in Section 3.2 each of $F_{\text{sum}}$ and $F_{\text{maxsum}}$ serve as a differentiable upper bound on $F_{\max}$ provided that $\Phi$ itself is differentiable. We will show that under certain mild assumptions these objective functions are essentially within a constant factor of each other. In view of the inequality $\sum_{y \neq y_i} \Phi\big(1 - \sum_{j=1}^{N} \alpha_j h_j(x_i, y_i, y)\big) \leq (c-1) \max_{y \neq y_i} \Phi\big(1 - \sum_{j=1}^{N} \alpha_j h_j(x_i, y_i, y)\big)$, the following inequalities relate $F_{\text{sum}}$, $F_{\max}$, and $F_{\text{maxsum}}$:

$$\frac{1}{c-1} F_{\text{sum}} \leq F_{\max} \leq F_{\text{maxsum}}. \tag{16}$$

Conversely, observe that due to the presence of the term $\sum_{j=1}^{N} \Lambda_j \alpha_j$ in all these objective functions, the domain of $\boldsymbol{\alpha}$ can be restricted to $B_+ = \{\boldsymbol{\alpha} \colon (0 \leq \boldsymbol{\alpha}) \wedge (\|\boldsymbol{\alpha}\|_1 \leq \Lambda)\}$ with the constant $\Lambda > 0$ depending only on $\Lambda_j$s. Then, the following inequality holds over $B_+$:

$$F_{\text{maxsum}} \leq \frac{e^{\Lambda}}{c-1} F_{\text{sum}}. \tag{17}$$

Indeed, if $\Phi = \exp$ or $\Phi(-x) = \log_2(1 + e^{-x})$, then $\Phi(x + b) \leq e^b \Phi(x)$ for any $b \geq 0$ and we can write

$$\Phi\Big(1 - \sum_{j=1}^{N} \alpha_j \rho_{h_j}(x_i, y_i)\Big)$$

$$= \Phi\Big(1 - \sum_{j=1}^{N} \alpha_j \frac{1}{c-1} \sum_{y \neq y_i} h_j(x_i, y_i, y) + \sum_{j=1}^{N} \alpha_j \frac{1}{c-1} \sum_{y \neq y_i} (h_j(x_i, y_i, y) - \rho_{h_j}(x_i, y_i))\Big)$$

$$\leq \frac{1}{c-1} \sum_{y \neq y_i} \Phi\Big(1 - \sum_{j=1}^{N} \alpha_j h_j(x_i, y_i, y) + 2\|\boldsymbol{\alpha}\|_1\Big)$$

$$\leq \frac{1}{c-1} \sum_{y \neq y_i} \Phi\Big(1 - \sum_{j=1}^{N} \alpha_j h_j(x_i, y_i, y)\Big) e^{2\|\boldsymbol{\alpha}\|_1},$$

where we used the convexity of $\Phi$ for the first inequality.

## E  MDeepBoostSum

### E.1  Direction

For any $j \in [1, N]$, $F_{\text{sum}}(\boldsymbol{\alpha}_{t-1} + \eta \mathbf{e}_j)$ is given by

$$F_{\text{sum}}(\boldsymbol{\alpha}_{t-1} + \eta \mathbf{e}_j) = \frac{1}{m} \sum_{i=1}^{m} \sum_{y \neq y_i} \Phi\Big(1 - f_{t-1}(x_i, y_i, y) - \eta h_j(x_i, y_i, y)\Big) + \sum_{j=1}^{N} \Lambda_j \alpha_{t-1,j} + \eta \Lambda_j. \tag{18}$$

Thus, for any $j \in [1, N]$, the directional derivative of $F_{\text{sum}}$ at $\boldsymbol{\alpha}_{t-1}$ along $\mathbf{e}_j$ can be expressed as follows in terms of $\epsilon_{t,j}$:

$$F'_{\text{sum}}(\boldsymbol{\alpha}_{t-1}, \mathbf{e}_j) = \frac{1}{m} \sum_{i=1}^{m} \sum_{y \neq y_i} (-\mathsf{h}_j(x_i, y_i, y)) \Phi'\left(1 - \mathsf{f}_{t-1}(x_i, y_i, y)\right) + \Lambda_j$$

$$= \frac{1}{m} \sum_{i=1}^{m} \sum_{y \neq y_i} (-\mathsf{h}_j(x_i, y_i, y)) \mathcal{D}_t(i, y) S_t + \Lambda_j = \frac{S_t}{m}(2\epsilon_{t,j} - 1) + \Lambda_j.$$

Thus, the direction $k$ selected at round $t$ is given by $k = \operatorname{argmin}_{j \in [1, N]} \epsilon_{t,j} + \frac{\Lambda_j m}{2 S_t}$.

## E.2 Step

Given the direction $\mathbf{e}_k$, the optimal step value $\eta$ is given by $\operatorname{argmin}_\eta F_{\text{sum}}(\boldsymbol{\alpha}_{t-1} + \eta \, \mathbf{e}_k)$. In the most general case, $\eta$ can be found via a line search or other numerical methods. In some special cases, we can derive a closed-form solution for the step by minimizing an upper bound on $F_{\text{sum}}(\boldsymbol{\alpha}_{t-1} + \eta \, \mathbf{e}_k)$.

Since for any $i \in [1, m]$ and $y \in \mathcal{Y}$, $\mathsf{h}_k(x_i, y_i, y) = \frac{1 + \mathsf{h}_k(x_i, y_i, y)}{2} \cdot (1) + \frac{1 - \mathsf{h}_k(x_i, y_i, y)}{2} \cdot (-1)$, by the convexity of $u \mapsto \Phi(1 - \eta u)$, the following holds for all $\eta \in \mathbb{R}$:

$$\Phi\left(1 - \mathsf{f}_{t-1}(x_i, y_i, y) - \eta \mathsf{h}_k(x_i, y_i, y)\right) \leq \frac{1 + \mathsf{h}_k(x_i, y_i, y)}{2} \Phi\left(1 - \mathsf{f}_{t-1}(x_i, y_i, y) - \eta\right)$$

$$+ \frac{1 - \mathsf{h}_k(x_i, y_i, y)}{2} \Phi\left(1 - \mathsf{f}_{t-1}(x_i, y_i, y) + \eta\right).$$

Thus, we can write

$$F(\boldsymbol{\alpha}_{t-1} + \eta \mathbf{e}_k) \leq \frac{1}{m} \sum_{i=1}^{m} \sum_{y \neq y_i} \frac{1 + \mathsf{h}_k(x_i, y_i, y)}{2} \Phi\left(1 - \mathsf{f}_{t-1}(x_i, y_i, y)) - \eta\right)$$

$$+ \frac{1}{m} \sum_{i=1}^{m} \sum_{y \neq y_i} \frac{1 - \mathsf{h}_k(x_i, y_i, y)}{2} \Phi\left(1 - \mathsf{f}_{t-1}(x_i, y_i, y)) + \eta\right) + \sum_{j=1}^{N} \alpha_{t-1,j} \Lambda_j + \eta \Lambda_k.$$

Let $J(\eta)$ denote that upper bound. We can select $\eta$ as the solution of $\min_{\eta + \alpha_{t-1,k} \geq 0} J(\eta)$. Since $J$ is convex, this defines a convex optimization problem.

### E.2.1 Exponential loss

In the case $\Phi = \exp$, $J(\eta)$ can be expressed as follows:

$$J(\eta) = \frac{1}{m} \sum_{i=1}^{m} \sum_{y \neq y_i} \frac{1 + \mathsf{h}_k(x_i, y_i, y)}{2} e^{1 - \mathsf{f}_{t-1}(x_i, y_i, y)} e^{-\eta}$$

$$+ \frac{1}{m} \sum_{i=1}^{m} \sum_{y \neq y_i} \frac{1 - \mathsf{h}_k(x_i, y_i, y)}{2} e^{1 - \mathsf{f}_{t-1}(x_i, y_i, y)} e^{\eta} + \sum_{j=1}^{N} \alpha_{t-1,j} \Lambda_j + \eta \Lambda_k,$$

with $e^{1 - \mathsf{f}_{t-1}(x_i, y_i, y)} = \Phi'(1 - \mathsf{f}_{t-1}(x_i, y_i, y)) = S_t \mathcal{D}_t(i, y)$. Thus, $J(\eta)$ can be rewritten as

$$J(\eta) = \frac{1}{m} \sum_{i=1}^{m} \sum_{y \neq y_i} \mathcal{D}_t(i, y) S_t e^{-\eta \mathsf{h}_k(x_i, y_i, y)} + \Lambda_k \eta$$

$$= (1 - \epsilon_{t,k}) \frac{S_t}{m} e^{-\eta} + \epsilon_{t,k} \frac{S_t}{m} e^{\eta} + \sum_{j=1}^{N} \alpha_{t-1,j} \Lambda_j + \eta \Lambda_k.$$

Introducing a Lagrange variable $\mu \geq 0$, the Lagrangian associated to the convex optimization problem $\min_{\eta + \alpha_{t-1,k} \geq 0} J(\eta)$ can be written as follows:

$$L(\eta, \mu) = J(\eta) - \mu(\eta + \alpha_{t-1,k}) \quad \text{with} \quad \nabla_\eta L(\eta, \mu) = J'(\eta) - \mu.$$

By the KKT conditions, at the solution $(\eta^*, \mu^*)$, $J'(\eta^*) = \mu^*$ and $\mu^*(\eta^* + \alpha_{t-1,k}) = 0$. Thus, either $(\mu^* > 0) \Leftrightarrow (J'(\eta^*) > 0)$ and $\eta^* = -\alpha_{t-1,k}$, or $\mu^* = 0$ and $\eta^*$ is solution of the equation $J'(\eta^*) = 0$.

The condition $J'(\eta^*) > 0$ for $\eta^* = -\alpha_{t-1,k}$ can be rewritten as

$$-(1 - \epsilon_{t,k})\frac{S_t}{m}e^{\alpha_{t-1,k}} + \epsilon_{t,k}\frac{S_t}{m}e^{-\alpha_{t-1,k}} + \Lambda_k > 0 \Leftrightarrow (1 - \epsilon_{t,k})e^{\alpha_{t-1,k}} - \epsilon_{t,k}e^{-\alpha_{t-1,k}} < \frac{\Lambda_k m}{S_t}.$$

$J'(\eta) = 0$ can be written as the second-degree equation $e^{2\eta} + \frac{\Lambda_k m}{S_t \epsilon_{t,k}}e^{\eta} - \frac{1 - \epsilon_{t,k}}{\epsilon_{t,k}}$, which admits the solution

$$e^{\eta} = -\frac{\Lambda_k m}{2 S_t \epsilon_{t,k}} + \sqrt{\left[\frac{\Lambda_k m}{2 S_t \epsilon_{t,k}}\right]^2 + \frac{1 - \epsilon_{t,k}}{\epsilon_{t,k}}} \Leftrightarrow \eta = \log\left[-\frac{\Lambda_k m}{2 S_t \epsilon_{t,k}} + \sqrt{\left[\frac{\Lambda_k m}{2 S_t \epsilon_{t,k}}\right]^2 + \frac{1 - \epsilon_{t,k}}{\epsilon_{t,k}}}\right].$$

### E.2.2   Logistic loss

In the case of the logistic loss, for any $u \in \mathbb{R}$, $\Phi(-u) = \log_2(1 + e^{-u})$ and $\Phi'(-u) = \frac{1}{\log 2}\frac{1}{(1+e^u)}$. To determine the step size, we use the following general upper bound:

$$\Phi(-u - v) - \Phi(-u) = \log_2\left[\frac{1 + e^{-u} + e^{-u-v} - e^{-u}}{1 + e^{-u}}\right]$$

$$= \log_2\left[1 + \frac{e^{-v} - 1}{1 + e^u}\right] \le \frac{e^{-v} - 1}{(\log 2)(1 + e^u)} = \Phi'(-u)(e^{-v} - 1).$$

Thus, we can write

$$F(\boldsymbol{\alpha}_{t-1} + \eta\mathbf{e}_k) - F(\boldsymbol{\alpha}_{t-1}) \le \frac{1}{m}\sum_{i=1}^{m}\sum_{y \ne y_i} \Phi'(1 - \mathsf{f}_{t-1}(x_i, y_i, y))(e^{-\eta\mathsf{h}_k(x_i, y_i, y)} - 1) + \Lambda_k \eta$$

$$= \frac{1}{m}\sum_{i=1}^{m}\sum_{y \ne y_i} \mathcal{D}_t(i, y)S_t(e^{-\eta\mathsf{h}_k(x_i, y_i, y)} - 1) + \Lambda_k \eta.$$

To determine $\eta$, we can minimize this upper bound, or equivalently the following

$$\frac{1}{m}\sum_{i=1}^{m}\sum_{y \ne y_i} \mathcal{D}_t(i, y)S_t e^{-\eta\mathsf{h}_k(x_i, y_i, y)} + \Lambda_k \eta.$$

This expression is syntactically the same as (18) in the case of the exponential loss (modulo a term not depending on $\eta$) with only the distribution weights $\mathcal{D}_t(i, y)$ and $S_t$ being different. Thus, we obtain immediately the same expressions for the step size in the case of the logistic loss but with $S_t = \sum_{i=1}^{m} \frac{1}{1 + e^{\mathsf{f}_{t-1}(x_i, y_i, y) - 1}}$ and $\mathcal{D}_t(i, y) = \frac{1}{S_t}\frac{1}{1 + e^{\mathsf{f}_{t-1}(x_i, y_i, y) - 1}}$.

## F   MDeepBoostMaxSum

MDeepBoostMaxSum algorithm is derived by application of coordinate descent to $F_{\text{maxsum}}$ objective function. Below we provide explicit expressions for the direction and step of the descent. Figure 2 gives the pseudocode of the MDeepBoostMaxSum algorithm.

### F.1   Direction

For any $j \in [1, N]$, $F_{\text{maxsum}}(\boldsymbol{\alpha}_{t-1} + \eta\mathbf{e}_j)$ is given by

$$F_{\text{maxsum}}(\boldsymbol{\alpha}_{t-1} + \eta\mathbf{e}_j) = \frac{1}{m}\sum_{i=1}^{m} \Phi\left(1 - \sum_{j=1}^{N}\alpha_{t-1,j}\rho_{h_j}(x_i, y_i) - \eta\rho_{h_j}(x_i, y_i)\right) + \sum_{j=1}^{N}\Lambda_j\alpha_{t-1,j} + \eta\Lambda_j.$$

$$\tag{19}$$

Thus, for any $j \in [1, N]$, the directional derivative of $F_{\text{maxsum}}$ at $\boldsymbol{\alpha}_{t-1}$ along $\mathbf{e}_j$ can be expressed as follows:

$$F'_{\text{maxsum}}(\boldsymbol{\alpha}_{t-1}, \mathbf{e}_j) = \frac{1}{m} \sum_{i=1}^{m} (-\rho_{h_j}(x_i, y_i)) \Phi'\left(1 - \sum_{j=1}^{N} \alpha_{t-1,j} \rho_{h_j}(x_i, y_i)\right) + \Lambda_j$$

$$= \frac{1}{m} \sum_{i=1}^{m} (-\rho_{h_j}(x_i, y_i)) \mathcal{D}_t(i) S_t + \Lambda_j = \frac{S_t}{m}(2\epsilon_{t,j} - 1) + \Lambda_j,$$

where for any $t \in [1, T]$, we denote by $\mathcal{D}_t$ the distribution over $[1, m]$ defined for all $i \in [1, m]$ by

$$\mathcal{D}_t(i) = \frac{\Phi'\left(1 - \sum_{j=1}^{N} \alpha_{t-1,j} \rho_{h_j}(x_i, y_i)\right)}{S_t}, \tag{20}$$

with normalization factor $S_t = \sum_{i=1}^{m} \Phi'(1 - \sum_{j=1}^{N} \alpha_{t-1,j} \rho_{h_j}(x_i, y_i))$. For any $j \in [1, N]$ and $s \in [1, T]$, we also define the weighted error $\epsilon_{s,j}$ as follows:

$$\epsilon_{s,j} = \frac{1}{2}\left[1 - \operatorname*{E}_{i \sim \mathcal{D}_s}\left[\rho_{h_s}(x_i, y_i)\right]\right]. \tag{21}$$

Thus, the direction $k$ selected by MDeepBoostMaxSum at round $t$ is given by $k = \operatorname{argmin}_{j \in [1,N]} \epsilon_{t,j} + \frac{\Lambda_j m}{2S_t}$.

## F.2 Step

Given the direction $\mathbf{e}_k$, the optimal step value $\eta$ is given by $\operatorname{argmin}_\eta F_{\text{maxsum}}(\boldsymbol{\alpha}_{t-1} + \eta \mathbf{e}_k)$. As in the case of MDeepBoostSum, for the most general $\Phi$, $\eta$ can be found via a line search or other numerical methods and in some special cases, we can derive a closed-form solution for the step by minimizing an upper bound on $F_{\text{maxsum}}(\boldsymbol{\alpha}_{t-1} + \eta \mathbf{e}_k)$.

Following the same convexity argument as in Section E.2, we can write

$$F(\boldsymbol{\alpha}_{t-1} + \eta \mathbf{e}_k) \leq \frac{1}{m} \sum_{i=1}^{m} \frac{1 + \rho_{h_k}(x_i, y_i)}{2} \Phi\left(1 - \sum_{j=1}^{N} \alpha_{t-1,j} \rho_{h_j}(x_i, y_i) - \eta\right)$$

$$+ \frac{1}{m} \sum_{i=1}^{m} \frac{1 - \rho_{h_k}(x_i, y_i)}{2} \Phi\left(1 - \sum_{j=1}^{N} \alpha_{t-1,j} \rho_{h_j}(x_i, y_i) + \eta\right)$$

$$+ \sum_{j=1}^{N} \alpha_{t-1,j} \Lambda_j + \eta \Lambda_k.$$

Let $J(\eta)$ denote that upper bound. We now examine solutions of the convex optimization problem $\min_{\eta + \alpha_{t-1,k} \geq 0} J(\eta)$ in the case of exponential and logistic loss.

### F.2.1 Exponential loss

In the case $\Phi = \exp$, arguing as in Section E.2.1, $J(\eta)$ can be expressed as follows:

$$J(\eta) = (1 - \epsilon_{t,k})\frac{S_t}{m} e^{-\eta} + \epsilon_{t,k}\frac{S_t}{m} e^{\eta} + \sum_{j=1}^{N} \alpha_{t-1,j}\Lambda_j + \eta\Lambda_k.$$

and the solution of the optimization problem $\min_{\eta + \alpha_{t-1,k} \geq 0} J(\eta)$ is given by $\eta^* = -\alpha_{t-1,k}$ if

$$(1 - \epsilon_{t,k})e^{\alpha_{t-1,k}} - \epsilon_{t,k}e^{-\alpha_{t-1,k}} < \frac{\Lambda_k m}{S_t}.$$

Otherwise, the solution is

$$\eta^* = \log\left[-\frac{\Lambda_k m}{2S_t \epsilon_{t,k}} + \sqrt{\left[\frac{\Lambda_k m}{2S_t \epsilon_{t,k}}\right]^2 + \frac{1 - \epsilon_{t,k}}{\epsilon_{t,k}}}\right].$$

MDEEPBOOSTMAXSUM($S = ((x_1, y_1), \ldots, (x_m, y_m))$)
1   **for** $i \leftarrow 1$ **to** $m$ **do**
2       $D_1(i) \leftarrow \frac{1}{m}$
3   **for** $t \leftarrow 1$ **to** $T$ **do**
4       $k \leftarrow \underset{j \in [1,N]}{\operatorname{argmin}} \; \epsilon_{t,j} + \dfrac{\Lambda_j m}{2 S_t}$
5       **if** $\left( (1 - \epsilon_{t,k}) e^{\alpha_{t-1,k}} - \epsilon_{t,k} e^{-\alpha_{t-1,k}} < \frac{\Lambda_k m}{S_t} \right)$ **then**
6           $\eta_t \leftarrow -\alpha_{t-1,k}$
7       **else** $\eta_t \leftarrow \log \left[ -\frac{\Lambda_k m}{2\epsilon_t S_t} + \sqrt{\left[ \frac{\Lambda_k m}{2\epsilon_t S_t} \right]^2 + \frac{1 - \epsilon_t}{\epsilon_t}} \; \right]$
8       $\boldsymbol{\alpha}_t \leftarrow \boldsymbol{\alpha}_{t-1} + \eta_t \mathbf{e}_k$
9       $S_{t+1} \leftarrow \sum_{i=1}^{m} \Phi'\big(1 - \sum_{j=1}^{N} \alpha_{t,j} \rho_{h_j}(x_i, y_i)\big)$
10      **for** $i \leftarrow 1$ **to** $m$ **do**
11          $D_{t+1}(i) \leftarrow \dfrac{\Phi'\left(1 - \sum_{j=1}^{N} \alpha_{t,j} \rho_{h_j}(x_i, y_i)\right)}{S_{t+1}}$
12  $f \leftarrow \sum_{j=1}^{N} \alpha_{t,j} h_j$
13  **return** $f$

Figure 2: Pseudocode of the MDeepBoostMaxSum algorithm for both the exponential loss and the logistic loss. The expression of the weighted error $\epsilon_{t,j}$ is given in (21). In the generic case of a surrogate loss $\Phi$ different from the exponential or logistic losses, $\eta_t$ is found instead via a line search or other numerical methods from $\eta_t = \operatorname{argmax}_\eta F_{\text{maxsum}}(\boldsymbol{\alpha}_{t-1} + \eta \mathbf{e}_k)$.

### F.2.2   Logistic loss

In the case of the logistic loss, we can argue as in Section E.2.2. In particular, we can write

$$F(\boldsymbol{\alpha}_{t-1} + \eta \mathbf{e}_k) - F(\boldsymbol{\alpha}_{t-1}) \leq \frac{1}{m} \sum_{i=1}^{m} \mathcal{D}_t(i) S_t (e^{-\eta \rho_{h_k}(x_i, y_i)} - 1) + \Lambda_k \eta.$$

As in the case of MDeepBoostSum algorithm with logistic loss, minimizing this upper bound, results in the same expressions for the step size $eta$ as in the case of the exponential loss but with normalization factor $S_t = \sum_{i=1}^{m} \left( 1 + e^{\sum_{j=1}^{N} \alpha_{t-1,j} \rho_{h_j}(x_i, y_i) - 1} \right)^{-1}$ and probability distribution $\mathcal{D}_t(i) = \frac{1}{S_t} \left( 1 + e^{\sum_{j=1}^{N} \alpha_{t-1,j} \rho_{h_j}(x_i, y_i) - 1} \right)^{-1}$.

## G   MDeepBoostCompSum

In this section, we describe the details of the MDeepBoostCompSum algorithm which consists of the application of coordinate descent to the $F_{\text{compsum}}$ objective function. Note that, in general, $F_{\text{compsum}}$ needs not be a convex function. However, in the important special case where $\Phi$ is the logistic function the objective does indeed define a convex optimization problem. Under this assumption, we show that the resulting algorithm is identical to the MDeepBoostSum algorithm with only the distribution weights $\mathcal{D}_t(i, y)$ and $S_t$ being different.

### G.1   Direction

For any $j \in [1, N]$, $F_{\text{compsum}}(\boldsymbol{\alpha}_{t-1} + \eta \mathbf{e}_j)$ is given by

$$F_{\text{sum}}(\boldsymbol{\alpha}_{t-1} + \eta \mathbf{e}_j) = \frac{1}{m} \sum_{i=1}^{m} \Phi_1 \Big( \sum_{y \neq y_i} \Phi_2 \Big( 1 - \mathsf{f}_{t-1}(x_i, y_i, y) - \eta \mathsf{h}_j(x_i, y_i, y) \Big) \Big) + \sum_{j=1}^{N} \Lambda_j \alpha_{t-1,j} + \eta \Lambda_j.$$
(22)

Thus, for any $j \in [1, N]$, the directional derivative of $F_{\text{compsum}}$ at $\boldsymbol{\alpha}_{t-1}$ along $\mathbf{e}_j$ can be expressed as follows in terms of $\epsilon_{t,j}$:

$$\frac{1}{m}\sum_{i=1}^{m}\Phi_1'\Big(\sum_{y\neq y_i}\Phi_2\big(1-\mathsf{f}_{t-1}(x_i,y_i,y)\big)\Big)\sum_{y\neq y_i}(-\mathsf{h}_j(x_i,y_i,y))\Phi_2'\big(1-\mathsf{f}_{t-1}(x_i,y_i,y)\big)+\Lambda_j$$

$$=\frac{1}{m}\sum_{i=1}^{m}\sum_{y\neq y_i}(-\mathsf{h}_j(x_i,y_i,y))\mathcal{D}_t(i,y)S_t+\Lambda_j=\frac{S_t}{m}(2\epsilon_{t,j}-1)+\Lambda_j,$$

where for any $t \in [1, T]$, we denote by $\mathcal{D}_t$ the distribution over $[1, m]$ defined for all $i \in [1, m]$ and all $y \in \mathcal{Y}$ such that $y \neq y_i$ by

$$\mathcal{D}_t(i,y)=\frac{\Phi_1'\Big(\sum_{y\neq y_i}\Phi_2\big(1-\mathsf{f}_{t-1}(x_i,y_i,y)\big)\Big)\Phi_2'\big(1-\mathsf{f}_{t-1}(x_i,y_i,y)\big)}{S_t}, \tag{23}$$

with normalization factor $S_t = \sum_{i=1}^{m}\Phi_1'\Big(\sum_{y\neq y_i}\Phi_2\big(1-\mathsf{f}_{t-1}(x_i,y_i,y)\big)\Big)\Phi_2'\big(1-\mathsf{f}_{t-1}(x_i,y_i,y)\big)$. For any $j \in [1, N]$ and $s \in [1, T]$, we also define the weighted error $\epsilon_{s,j}$ as follows:

$$\epsilon_{s,j}=\frac{1}{2}\Big[1-\mathop{\mathrm{E}}_{(i,y)\sim\mathcal{D}_s}\big[\mathsf{h}_s(x_i,y_i,y)\big]\Big]. \tag{24}$$

Thus, the direction $k$ selected at round $t$ is given by $k = \operatorname{argmin}_{j\in[1,N]}\epsilon_{t,j} + \frac{\Lambda_j m}{2S_t}$.

## G.2 Step

Given the direction $\mathbf{e}_k$, the optimal step value $\eta$ is given by $\operatorname{argmin}_\eta F_{\text{compsum}}(\boldsymbol{\alpha}_{t-1} + \eta\,\mathbf{e}_k)$. The most general case can be handled via a line search or other numerical methods. However, we recall that in this most general case objective of our problem need not be convex. In what follows, we assume that $\Phi$ is the logistic loss function and show that for resulting convex optimization problem, step can be chosen in the same way as for MDeepBoostSum algorithm of E.2.1.

To simplify the notation, for a fixed $i$, let $u(y) = 1 - \mathsf{f}_{t-1}(x_i, y_i, y)$ and $v(y) = -\eta\mathsf{h}_j(x_i, y_i, y)$. Then we can write

$$\Phi_1\Big(\sum_{y\neq y_i}\Phi_2(u(y)+v(y))\Big)-\Phi_1\Big(\sum_{y\neq y_i}\Phi_2(u(y))\Big)=\log_2\left[1+\frac{\sum_{y\neq y_i}e^{u(y)}(e^{v(y)}-1)}{1+\sum_{y\neq y_i}e^{u(y)}}\right]$$

$$\leq\frac{\sum_{y\neq y_i}e^{u(y)}(e^{v(y)}-1)}{1+\sum_{y\neq y_i}e^{u(y)}}.$$

This bound is precisely $S_t\sum_{y\neq y_i}\mathcal{D}_t(i,y)(e^{-\eta\mathsf{h}_k(x_i,y_i,y)}-1)$ and we can write

$$F(\boldsymbol{\alpha}_{t-1}+\eta\mathbf{e}_k)-F(\boldsymbol{\alpha}_{t-1})\leq\frac{1}{m}\sum_{i=1}^{m}\sum_{y\neq y_i}\mathcal{D}_t(i,y)S_t(e^{-\eta\mathsf{h}_k(x_i,y_i,y)}-1)+\Lambda_k\eta.$$

To determine $\eta$, we can minimize this upper bound, or equivalently the following

$$\frac{1}{m}\sum_{i=1}^{m}\sum_{y\neq y_i}\mathcal{D}_t(i,y)S_te^{-\eta\mathsf{h}_k(x_i,y_i,y)}+\Lambda_k\eta.$$

Arguing as in Section E.2.2, one can show that minimizing yields $\eta^* = -\alpha_{t-1,k}$ when

$$(1-\epsilon_{t,k})e^{\alpha_{t-1,k}}-\epsilon_{t,k}e^{-\alpha_{t-1,k}}<\frac{\Lambda_k m}{S_t}.$$

and otherwise

$$\eta=\log\left[-\frac{\Lambda_k m}{2S_t\epsilon_{t,k}}+\sqrt{\Big[\frac{\Lambda_k m}{2S_t\epsilon_{t,k}}\Big]^2+\frac{1-\epsilon_{t,k}}{\epsilon_{t,k}}}\right].$$

This shows that in the case of the logistic loss MDeepBoostCompSum is identical to MDeepBoost-Sum algorithm with only the distribution weights $\mathcal{D}_t(i,y)$ and $S_t$ being different.

# H MDeepBoostMax

MDeepBoostMax algorithm is derived by application of coordinate descent to $F_{\max}$ objective function.

## H.1 Direction and step

For simplicity we will assume that $\Phi$ is a twice differentiable strictly convex function. This is a mild assumption which holds for both exponential and logistic loss functions. Let $\boldsymbol{\alpha}_t = (\alpha_{t,1}, \ldots, \alpha_{t,N})^\top$ denote the vector obtained after $t \geq 1$ iterations and let $\boldsymbol{\alpha}_0 = \mathbf{0}$. Let $\mathbf{e}_k$ denote the $k$th unit vector in $\mathbb{R}^N$, $k \in [1, N]$. The direction $\mathbf{e}_k$ and the step $\eta$ selected at the $t$th round are those minimizing $F_{\max}(\boldsymbol{\alpha}_{t-1} + \eta \mathbf{e}_k)$, that is

$$F_{\max}(\boldsymbol{\alpha}_{t-1} + \eta \mathbf{e}_k) = \frac{1}{m} \sum_{i=1}^m \max_{y \neq y_i} \Phi\Big(1 - \mathsf{f}_{t-1}(x_i, y_i, y) - \eta \mathsf{h}_k(x_i, y_i, y)\Big)$$

$$+ \sum_{j \neq k} \Lambda_j \alpha_{t-1,j} + \Lambda_k \alpha_{t-1,k} + \eta. \quad (25)$$

We follow the definition of maximum coordinate descent for non-differentiable convex functions Cortes et al. [2014]. In view of that, at each iteration $t \geq 1$, the direction $\mathbf{e}_k$ selected by coordinate descent with maximum descent coordinate is $k = \operatorname{argmax}_{j \in [1,N]} |\delta F_{\max}(\boldsymbol{\alpha}_{t-1}, \mathbf{e}_j)|$, where $\delta F_{\max}(\boldsymbol{\alpha}_{t-1}, \mathbf{e}_j)$ is the element of the sub-gradient of $F_{\max}$ along $\mathbf{e}_j$ that is the closest to 0.

For any $i \in [1, m]$, let $\mathcal{Y}_{t,i} = \operatorname{argmax}_{y \neq y_i} \Phi(1 - \mathsf{f}_{t-1}(x_i, y_i, y))$ and $\phi_i$ be defined by $\phi_i(\boldsymbol{\alpha}_{t-1} + \eta \mathbf{e}_j) = \max_{y \neq y_i} \Phi\big(1 - \mathsf{f}_{t-1}(x_i, y_i, y) - \eta \mathsf{h}_j(x_i, y_i, y)\big)$. Then, since the right-derivative of $\phi_i$ at $\boldsymbol{\alpha}_{t-1}$ along the direction $\mathbf{e}_j$ is the largest element of the sub-differential of $\phi_i$ at $\boldsymbol{\alpha}_{t-1}$ and sub-differential of $\phi_i$ is a convex hull of $\frac{d}{d\eta} \Phi\big(1 - \mathsf{f}_{t-1}(x_i, y_i, y) - \eta \mathsf{h}_j(x_i, y_i, y)\big)\big|_{\eta=0}$ for $y \in \mathcal{Y}_{t,i}$, we can write

$$\phi'_{i,+}(\boldsymbol{\alpha}_{t-1}, \mathbf{e}_j) = \max_{y \in \mathcal{Y}_{t,i}} -h_j(x_i, y_i, y) \Phi'\big(1 - \mathsf{f}_{t-1}(x_i, y_i, y)\big) = \Phi'_{t-1,i} \max_{y \in \mathcal{Y}_{t,i}} \{-h_j(x_i, y_i, y)\},$$

where $\Phi'_{t-1,i} = \max_{y \neq y_i} \Phi'\big(1 - \mathsf{f}_{t-1}(x_i, y_i, y)\big)$. The last equality is a consequence of the fact that $\Phi'\big(1 - \mathsf{f}_{t-1}(x_i, y_i, y)\big) = \max_{y \neq y_i} \Phi'\big(1 - \mathsf{f}_{t-1}(x_i, y_i, y)\big)$ for all $y \in \mathcal{Y}_{t,i}$ and hence can be factored out of $\max_{y \in \mathcal{Y}_{t,i}} -h_j(x_i, y_i, y) \Phi'\big(1 - \mathsf{f}_{t-1}(x_i, y_i, y)\big)$. This is indeed the case since for a twice differentiable strictly convex function $\Phi$, $\Phi'' > 0$ and $\Phi'$ is strictly increasing. Combining this with monotonicity of $\Phi$ we have

$$\mathcal{Y}_{t,i} = \operatorname*{argmax}_{y \neq y_i} \Phi(1 - \mathsf{f}_{t-1}(x_i, y_i, y)) = \operatorname*{argmax}_{y \neq y_i} \Phi'(1 - \mathsf{f}_{t-1}(x_i, y_i, y)).$$

Similarly, we can write

$$\phi'_{i,-}(\boldsymbol{\alpha}_{t-1}, \mathbf{e}_j) = \min_{y \in \mathcal{Y}_{t,i}} -h_j(x_i, y_i, y) \Phi'\big(1 - \mathsf{f}_{t-1}(x_i, y_i, y)\big) = \Phi'_{t-1,i} \min_{y \in \mathcal{Y}_i} \{-h_j(x_i, y_i, y)\}.$$

In view of these identities, the right- and left-derivatives of $F$ along $\mathbf{e}_j$ are given by

$$F'_{\max,+}(\boldsymbol{\alpha}_{t-1}, \mathbf{e}_j) = \frac{1}{m} \sum_{i=1}^m \Phi'_{t-1,i} \max_{y \in \mathcal{Y}_{t,i}} \{-h_j(x_i, y_i, y)\} + \Lambda_j,$$

$$F'_{\max,-}(\boldsymbol{\alpha}_{t-1}, \mathbf{e}_j) = \frac{1}{m} \sum_{i=1}^m \Phi'_{t-1,i} \min_{y \in \mathcal{Y}_{t,i}} \{-h_j(x_i, y_i, y)\} + \Lambda_j.$$

For any $t \in [1, T]$, we denote by $\mathcal{D}_t$ the distribution defined by

$$\mathcal{D}_t(i) = \frac{\max_{y \neq y_i} \Phi'\big(1 - \mathsf{f}_{t-1}(x_i, y_i, y)\big)}{S_t}, \quad (26)$$

where $S_t$ is a normalization factor, $S_t = \sum_{i=1}^m \max_{y \neq y_i} \Phi'\big(1 - \mathsf{f}_{t-1}(x_i, y_i, y)\big)$. For any $s \in [1, T]$ and $j \in [1, N]$, we denote by $\epsilon_{s,j}^+$ and $\epsilon_{s,j}^-$ the following weighted errors of hypothesis $h_j$ for the distribution $\mathcal{D}_s$, for $s \in [1, T]$:

$$\epsilon_{s,j}^+ = \frac{1}{2}\Big[1 - \operatorname*{E}_{i \sim \mathcal{D}_s} \big[ \min_{y \in \mathcal{Y}_{s,i}} \mathsf{h}_j(x_i, y_i, y)\big]\Big] \qquad \epsilon_{s,j}^- = \frac{1}{2}\Big[1 - \operatorname*{E}_{i \sim \mathcal{D}_s} \big[ \max_{y \in \mathcal{Y}_{s,i}} \mathsf{h}_j(x_i, y_i, y)\big]\Big]. \quad (27)$$

```
MDEEPBOOSTMAX(S = ((x_1, y_1), . . . , (x_m, y_m)))
 1   for i ← 1 to m do
 2        D_1(i) ← 1/m
 3        𝒴_{1,i} ← 𝒴 − {y_i}
 4   for t ← 1 to T do
 5        for j ← 1 to N do
 6             if 1/2 − ε_{t,j}^+ ≥ Λ_j m / 2S_t then
 7                  d_j ← S_t/m (2ε_{t,j}^+ − 1) + Λ_j
 8             elseif 1/2 − ε_{t,j}^- ≤ Λ_j m / 2S_t then
 9                  d_j ← S_t/m (2ε_{t,j}^- − 1) + Λ_j
10             else  d_j ← 0
11        k ← argmax |d_j|
                j∈[1,N]
12        η_t ← argmin F_max(α_{t−1} + η e_k)
                η≥−α_k
13        α_t ← α_{t−1} + η_t e_k
14        S_{t+1} ← Σ_{i=1}^m max_{y≠y_i} Φ'(1 − f_{t−1}(x_i, y_i, y))
15        for i ← 1 to m do
16             D_{t+1}(i) ← max_{y≠y_i} Φ'(1−f_{t−1}(x_i,y_i,y)) / S_{t+1}
17             𝒴_{t,i} ← argmax Φ(1 − f_{t−1}(x_i, y_i, y))
                         y≠y_i
18   f ← Σ_{j=1}^N α_{t,j} h_j
19   return f
```

Figure 3: Pseudocode of the MDeepBoostMax algorithm. The expression of the weighted errors $\epsilon_{t,j}^+$ and $\epsilon_{t,j}^-$ is given in (27) and $\eta_t$ is found via a line search or other numerical methods. Note that the active label sets $\mathcal{Y}_{t,i}$ are needed for finding $\epsilon_{t,j}^+$ and $\epsilon_{t,j}^-$.

Since $\Phi'_{t-1,i} = S_t \mathcal{D}_t(i)$, we can express the right- and left-derivative in terms of $\epsilon_{t,j}^+$ and $\epsilon_{t,j}^-$:

$$F'_{\max,+}(\boldsymbol{\alpha}_{t-1}, \mathbf{e}_j) = \frac{S_t}{m}[2\epsilon_{t,j}^+ - 1] + \Lambda_j$$

$$F'_{\max,-}(\boldsymbol{\alpha}_{t-1}, \mathbf{e}_j) = \frac{S_t}{m}[2\epsilon_{t,j}^- - 1] + \Lambda_j.$$

Therefore, we can write

$$\delta F_{\max}(\boldsymbol{\alpha}_{t-1}, \mathbf{e}_j) = \begin{cases} \frac{S_t}{m}[2\epsilon_{t,j}^+ - 1] + \Lambda_j & \text{if } \frac{1}{2} - \epsilon_{t,j}^+ \geq \frac{\Lambda_j m}{2S_t} \\ \frac{S_t}{m}[2\epsilon_{t,j}^- - 1] + \Lambda_j & \text{else if } \frac{1}{2} - \epsilon_{t,j}^- \leq \frac{\Lambda_j m}{2S_t} \\ 0 & \text{otherwise,} \end{cases} \tag{28}$$

and the direction $k$ selected at round $t$ is given by $k = \operatorname{argmax}_{j \in [1,N]} |\delta F_{\max}(\boldsymbol{\alpha}_{t-1}, \mathbf{e}_j)|$. Given the direction $\mathbf{e}_k$, the optimal step value $\eta$ is given by $\operatorname{argmin}_{\eta \geq -\alpha_k} F_{\max}(\boldsymbol{\alpha}_{t-1} + \eta \mathbf{e}_k)$. This is a convex optimization problem that can be solved via a line search or other numerical methods.

Figure 3 gives the pseudocode of the MDeepBoostMax algorithm.

Note that convergence guarantees of Theorem 2 do not apply to $F_{\max}$ objective since the presence of the max operator makes it non-differentiable. In the most general setting of non-differentiable continuous objective functions, it is possible to construct examples where coordinate descent algorithm will get "stuck" and never reach the global minimum. More precisely, for a non-differentiable convex objective function $F$ the set of points such that $\delta F(\boldsymbol{\alpha}^*, \mathbf{e}_j) = 0$ for all $j \in [1, N]$ and yet $F(\boldsymbol{\alpha}^*) > \min_{\boldsymbol{\alpha}} F(\boldsymbol{\alpha})$ may not be empty. One way to address this problem and prevent coordinate descent from being "stuck" is to randomize it. Namely, every time we reach a point where

$\delta F(\boldsymbol{\alpha}, \mathbf{e}_j) = 0$ for all $j \in [1, N]$ we perturb $\boldsymbol{\alpha}$ by a small random vector $\boldsymbol{\alpha}_0$ and restart coordinate descent procedure from $\boldsymbol{\alpha} + \boldsymbol{\alpha}_0$.

# I   Convergence of coordinate descent

**Theorem 2.** *Assume that $\Phi$ is twice differentiable and that $\Phi''(u) > 0$ for all $u \in \mathbb{R}$. Then, the projected coordinate descent algorithm applied to $F$ converges to the solution $\boldsymbol{\alpha}^*$ of the optimization $\max_{\boldsymbol{\alpha} \geq 0} F(\boldsymbol{\alpha})$ for $F = F_{sum}$, $F = F_{maxsum}$, or $F = F_{compsum}$. If additionally $\Phi$ is strongly convex over the path of the iterates $\boldsymbol{\alpha}_t$, then there exists $\tau > 0$ and $\gamma > 0$ such that for all $t > \tau$,*

$$F(\boldsymbol{\alpha}_{t+1}) - F(\boldsymbol{\alpha}^*) \leq (1 - \tfrac{1}{\gamma})(F(\boldsymbol{\alpha}_t) - F(\boldsymbol{\alpha}^*)). \tag{29}$$

*Proof.* We present the proof in the case $F = F_{\text{sum}}$, the proof for the other cases is similar. Let $\mathbf{H}$ be the matrix in $\mathbb{R}^{m(c-1) \times N}$ defined by $\mathbf{H}_{(i,c),j} = \mathsf{h}_j(x_i, y_i, y)$ for all $i \in [1, m]$, $y \neq y_i$, and $j \in [1, N]$, and let $\mathbf{e}_{(i,y)}$ be the $(i, y)$th unit vector in $\mathbb{R}^{m(c-1)}$. Then, for any $\boldsymbol{\alpha}$, $\mathbf{e}_{(i,y)}^\top \mathbf{H} \boldsymbol{\alpha} = \sum_{j=1}^N \alpha_j \mathsf{h}_j(x_i, y_i, y)$. Thus, we can write for any $\boldsymbol{\alpha} \in \mathbb{R}^N$,

$$F_{\text{sum}}(\boldsymbol{\alpha}) = G(\mathbf{H}\boldsymbol{\alpha}) + \Lambda^\top \boldsymbol{\alpha},$$

where $\Lambda = (\Lambda_1, \ldots, \Lambda_N)^\top$ and where $G$ is the function defined by

$$G(\mathbf{u}) = \frac{1}{m} \sum_{i=1}^m \sum_{y \neq y_i} \Phi(1 - \mathbf{e}_{(i,y)}^\top \mathbf{u}) = \frac{1}{m} \sum_{i=1}^m \sum_{y \neq y_i} \Phi(1 - u_{(i,y)}),$$

for all $\mathbf{u} \in \mathbb{R}^{m(c-1)}$ with $u_{(i,y)}$ its $(i, y)$th coordinate. $G$ is twice differentiable since $\Phi$ is and $\nabla^2 G(\mathbf{u})$ is a diagonal matrix with diagonal entries $\frac{1}{m} \Phi''(1 - u_{(i,y)}) > 0$ for all $i \in [1, m]$ and $y \neq y_i$. Thus, $\nabla^2 G(\mathbf{H}\boldsymbol{\alpha})$ is positive definite for all $\boldsymbol{\alpha}$. The conditions of Theorem 2.1 of [Luo and Tseng, 1992] are therefore satisfied for the optimization problem

$$\min_{\boldsymbol{\alpha} \geq 0} G(\mathbf{H}\boldsymbol{\alpha}) + \Lambda^\top \boldsymbol{\alpha},$$

thereby guaranteeing the convergence of the projected coordinate descent method applied to $F_{\text{sum}}$. If additionally $F$ is strongly convex over the sequence of $\boldsymbol{\alpha}_t$s, then, by the results of [Luo and Tseng, 1992][page 26], the inequality (10) holds for the projected coordinate method that we are using which selects the best direction at each round, as with the Gauss-Southwell method. $\square$

Note that the result holds under the weaker condition that $\Phi''(1 - \sum_{j=1}^N \alpha_j^* \mathsf{h}_j(x_i, y_i, y)) > 0$ instead of $\Phi''(u) > 0$ for all $u$, since the assumptions of Theorem 2.1 of [Luo and Tseng, 1992] are also satisfied in that case.

## J   Upper bound on Rademacher complexity

In this section we prove Eq. (13), an upper bound on the Rademacher complexity of $\mathcal{T}_n$. We have

$$\Re(\Pi_1(\mathcal{T}_n)) = \frac{1}{m} \mathop{\mathrm{E}}_{\boldsymbol{\sigma}} \left[ \sup_{\mathtt{t}, \mathbf{h}, y \in \mathcal{Y}} \sum_{i=1}^{m} \sigma_i \sum_{l \in \mathrm{Leaves}(\mathtt{t})} 1_{\mathtt{t}(x)=l}\, h_l(y) \right]$$

$$= \frac{1}{m} \mathop{\mathrm{E}}_{\boldsymbol{\sigma}} \left[ \sup_{\mathtt{t}} \sup_{\mathbf{h}, y \in \mathcal{Y}} \sum_{l \in \mathrm{Leaves}(\mathtt{t})} h_l(y) \sum_{i=1}^{m} \sigma_i 1_{x_i \in l} \right]$$

$$= \frac{1}{m} \mathop{\mathrm{E}}_{\boldsymbol{\sigma}} \left[ \sup_{\mathtt{t}} \sum_{l \in \mathrm{Leaves}(\mathtt{t})} \left[ \sum_{i=1}^{m} \sigma_i 1_{x_i \in l \in \mathrm{Leaves}(\mathtt{t})} \right]_+ \right] \qquad \text{(take } h_l(y) = 0 \text{ or } h_l(y) = 1)$$

$$\leq \frac{1}{m} \mathop{\mathrm{E}}_{\boldsymbol{\sigma}} \left[ \sup_{\mathtt{t}} \sum_{l \in \mathrm{Leaves}(\mathtt{t})} \left| \sum_{i=1}^{m} \sigma_i 1_{x_i \in l} \right| \right]$$

$$= \frac{1}{m} \mathop{\mathrm{E}}_{\boldsymbol{\sigma}} \left[ \sup_{\mathtt{t}, s_l \in \{+1, -1\}} \sum_{l \in \mathrm{Leaves}(\mathtt{t})} s_l \sum_{i=1}^{m} \sigma_i 1_{x_i \in l} \right]$$

$$= \frac{1}{m} \mathop{\mathrm{E}}_{\boldsymbol{\sigma}} \left[ \sup_{\mathtt{t}, s_l \in \{+1, -1\}} \sum_{i=1}^{m} \sigma_i \sum_{l \in \mathrm{Leaves}(\mathtt{t})} s_l 1_{x_i \in l} \right].$$

This last expression coincides with the Rademacher complexity of decision trees in the binary classification case returning a value in $\{+1, -1\}$. The VC-dimension of this family can be bounded by $(2n + 1) \log_2(d + 2)$ (see for example [Mohri et al., 2012]). Thus, by Massart's lemma, Eq. 13 follows.

## K   Relationship with other algorithms

The view of boosting as coordinate descent applied to an objective function was pointed out and studied in detail by several authors in the past [Friedman et al., 1998, Duffy and Helmbold, 1999, Mason et al., 1999, Collins et al., 2002].

As pointed out earlier, the objective function $F_{\max}$ is the tightest convex surrogate among those we discussed and has favorable $H$-consistency properties. However, we are not aware of any prior algorithms based on this objective function, even without regularization ($\Lambda_j = 0$ for all $j$). Similarly, the objective function $F_{\mathrm{maxsum}}$ leads to a very efficient training algorithm since it is based on base classifier margins $\rho_{h_i}$ that can all be pre-computed before training begins, but we are not aware of prior work based on that objective. Thus, the corresponding algorithms MDeepBoostMax and MDeepBoostMaxSum are both entirely new.

Certain special cases of our algorithms coincide with well-known multi-class classification algorithms from the literature. For $\Lambda_j = 0$, $j \in [1, N]$ and the exponential loss ($\Phi(-u) = \exp(-u)$), the MDeepBoostSum algorithm is equivalent to AdaBoost.MR [Freund and Schapire, 1997, Schapire and Singer, 1999], a multi-class version of AdaBoost. For $\Lambda_j = 0$, $j \in [1, N]$ and the logistic loss ($\Phi(-u) = \log_2(1 + \exp(-u + 1))$), the MDeepBoostCompSum algorithm is equivalent to additive multinomial logistic regression (i.e., a conditional maximum entropy model) where $\Phi_1(x) = \log(1 + x)$ and $\Phi_2(x) = \exp(x + 1)$) (see Friedman [2000]). For the same $\Phi$, $\Phi_1$ and $\Phi_2$, when $\lambda = 0$ and $\beta \neq 0$ the MDeepBoostCompSum algorithm is equivalent to the multi-class logistic regression algorithm with $L_1$-norm regularization studied by Duchi and Singer [2009].

Other existing multi-class classification algorithms are related to the ones we present here, but with key differences. For example, the MDeepBoostCompSum algorithm is similar to several algorithms of Zou et al. [2008], except that they do not use regularization and additionally require the consistency condition $\sum_{y \mathcal{Y}} f(x, y) = 0$. Bühlmann and Yu [2003] also describe a multi-class classifica-

tion algorithm based on boosting, except they reduce the problem to binary classification using a one-versus-all approach, use the square loss for $\Phi$, and do not use regularization.

In the special case of binary classification, our algorithms are of course related to the binary classification deep boosting [Cortes et al., 2014] and to several boosting algorithms introduced in the past [Freund and Schapire, 1997, Kivinen and Warmuth, 1999, Rätsch et al., 2001a, Rätsch and Warmuth, 2002, 2005, Warmuth et al., 2006] including boosting with $L_1$-norm regularization [Rätsch et al., 2001a] (see [Schapire and Freund, 2012] for a more extended list of references), as discussed by Cortes et al. [2014].

## L    Dataset statistics

Table 4: Dataset statistics.

| Data set | Classes | Examples | Features |
|---|---|---|---|
| abalone | 29 | 4177 | 8 |
| handwritten | 10 | 5620 | 64 |
| letters | 26 | 20000 | 16 |
| pageblocks | 5 | 5473 | 10 |
| pendigits | 10 | 10992 | 16 |
| satimage | 6 | 6435 | 36 |
| statlog | 7 | 2310 | 19 |
| yeast | 10 | 1484 | 8 |

## Footnotes

[3] The number $S(p,n)$ of $p$-tuples $\mathbf{N}$ with $|\mathbf{N}| = n$ is known to be precisely $\binom{p+n-1}{p-1}$.

[4]To select $n$ we consider $f(n) = ce^{-nu} + \sqrt{nv}$, where $u = \rho^2/8$ and $v = \log p/m$. Taking the derivative of $f$, setting it to zero and solving for $n$, we obtain $n = -\frac{1}{2u} W_{-1}(-\frac{v}{2c^2 u})$ where $W_{-1}$ is the second branch of the Lambert function (inverse of $x \mapsto xe^x$). Using the bound $-\log x \leq -W_{-1}(-x) \leq 2 \log x$ leads to the following choice of $n$: $n = \left\lceil -\frac{1}{2u} \log(\frac{v}{2c^2 u}) \right\rceil$.