[Reviews · NeurIPS 2014]

Submitted by Assigned_Reviewer_5

Summary of the paper:

The paper introduces a new class of multi-class boosting algorithms with base learners that are regularized based on their Rademacher complexities. The experiments show great improvement over other multi-class approaches in the literature and on the non-regularized version of the proposed algorithm.

Detailed remarks:

For the title: in general, the term "deep" in deep learning is used for saying that the representation is hierarchical in a compositional sense. Trees are obviously not deep in this sense, and so just because you are using large trees in boosting (a quite common setup in practice), it will not make the approach "deep". You still use a single "hidden layer" of trees, combined linearly, this is definitely a shallow architecture. Thus I strongly object to the title, it is misleading.

Lines 042-043: In practice, boosted trees almost always outperform boosted stumps \cite{CaNi06, Keg14}, and when the validated tree sizes are obtained by proper hyper-parameter optimization, they can be quite large, the same order as found in your experiments. Moreover, when boosting multi-class Hamming trees in AB.MH, \cite{Keg14} also found that on most of the data sets there is very little overfitting, basically one can boost trees of several tens of inner nodes for ten thousand iterations (see, for example, pendigits or letters, two sets on which most algorithms are tested), without increasing the test error. So, the statement of "boosting has been observed to overfit in practice", derived from 15 year-old papers, should be revised. Sometimes it overfits, sometimes it doesn't, and basically we don't know when it does and why it does when it does.

Lines 046-047: To my knowledge, the first paper proposing adaptive regularization of base classifiers is \cite{KeWa04}. The intuitive idea is the same and the final algorithm is not that different either (coefficients have to be shrunk by a quantity related to the empirical complexity of the weak classifier).

Although we cannot expect that a conference paper surveys all multiclass boosting algorithms, the paper should at least mention those that seem to be state of the art: AOSO \cite{SuReZh12}, ABC \cite{Li09,Li09a}, Gao-Koller's iterative weak learner in hinge-boost \cite{GaKo11}, and AB.MH with Hamming trees \cite{Keg14} (it seems to me that the trees in this latter are quite similar to those used in this submission).

The experimental setup.

I was honestly stunned reading this:

"We recorded the parameter tuple that had the lowest average error across all 10 runs, and this average error and the standard deviation of the error is reported in Table 1 and Table 2, along with the average number of trees and the average size of the trees in the ensembles."

You're validating on the test set, something that we teach to our students never to do. The consequence is twofold. First, I cannot compare the errors to those available in the literature. Some of the errors (e.g. on pendigit) looked suspiciously low, 3-4 times lower than I've ever seen, that's when I started to check your experimental setup. Second, the empirical comparison of the algorithms you tested is tainted. It is obvious that if you take an algorithm and you add hyper-parameters (the AB.MR -> L1 AB.MR -> MDeepBoostSum chain), the minimum test error can only decrease. The ballpark range of the "improvements" is very much in line with this view: you simply harvested the fact that the minimum of a larger sample is smaller than the minimum of a smaller sample, even if they come from the same distribution.

Now, I know that this seems like a detail for a theoretician, but for people using these algorithms what you are claiming is important. We have tested a lot of ways of regularizing the weak learners up to about ten years ago (you referred to some of the works), it never worked, more precisely, we didn't seem to need it. There were some indications that it could help on small data sets \cite{KeWa04}, but, exactly because of the small size of the sets, results were inconclusive. If you now claim that it is not the case, the experimental validation has to be rock solid.

My suggestion is that you make an attempt to redo the experiments doing proper double cross validation during the rebuttal period, and show us the new results. If they are non-conclusive (that is, the regularized version doesn't beat the standard algorithm), I would say the paper could still be accepted, but the message has to be altered to something like "here is an interesting-looking algorithm with some strong theoretical justifications, but regularization doesn't work".

Providing an open source implementation of the algorithm would be a great way to make the experiments reproducible and to let people use the proposed techniques and to build on them.

Pseudocode: for those who would like to implement the method starting from the pseudocode, it would be helpful to point towards the definitions of the quantities in the caption: \Lambda and S_t for t = 1 are undefined.

@inproceedings{KeWa04,
Address = {Vancouver, Canada},
Author = {K\'{e}gl, B. and Wang, L.},
Booktitle = {Advances in Neural Information Processing Systems},
Pages = {665--672},
Publisher = {The MIT Press},
Title = {Boosting on manifolds: adaptive regularization of base classifiers},
Volume = {17},
Year = {2004}}

@inproceedings{CaNi06,
Author = {Caruana, R. and Niculescu-Mizil, A.},
Booktitle = {Proceedings of the 23rd International Conference on Machine Learning},
Pages = {161--168},
Title = {An Empirical Comparison of Supervised Learning Algorithms},
Year = {2006}}

@inproceedings{KeBu09,
Address = {Montreal, Canada},
Author = {K\'{e}gl, B. and Busa-Fekete, R.},
Booktitle = {International Conference on Machine Learning},
Pages = {497--504},
Title = {Boosting products of base classifiers},
Volume = {26},
Year = {2009}}

@inproceedings{Li09,
Author = {Li, P.},
Booktitle = {International Conference on Machine Learning},
Title = {{ABC}-{B}oost: Adaptive Base Class Boost for Multi-class Classification},
Year = {2009}}

@techreport{Li09a,
Author = {Li, P.},
Institution = {Arxiv preprint},
Number = {arXiv:0908.4144},
Title = {{ABC-LogitBoost} for Multi-class Classification},
Year = {2009}}

@inproceedings{GaKo11,
Author = {Gao, T. and Koller, D.},
Booktitle = {International Conference on Machine Learning},
Title = {Multiclass boosting with hinge loss based on output coding},
Year = {2011}}

@inproceedings{SuReZh12,
Author = {Sun, P. and Reid, M. D. and Zhou, J.},
Booktitle = {International Conference on Machine Learning (ICML)},
Title = {{AOSO-LogitBoost}: Adaptive One-Vs-One {LogitBoost} for Multi-Class Problem},
Year = {2012}}

@inproceedings{Keg14,
Abstract = { We train vector-valued decision trees within the framework of AdaBoost.MH. The key element of the method is a vector-valued decision stump, factorized into an input-independent vector of length $K$ and label-independent scalar classifier.},
Author = {K\'{e}gl, B.},
Booktitle = {International Conference on Learning Representations},
Title = {The return of {AdaBoost.MH}: multi-class {H}amming trees},
Url = {http://arxiv.org/abs/1312.6086},
Year = {2014}}

----

I looked at the new experimental results and indeed they are more reasonable now. As expected, errors increased significantly (eg., on letters and pendigits the errors increased by 6-7 standard deviations). In this light your answer

"1) While it is of course possible to overestimate the performance of a learning algorithm by optimizing hyperparameters on the test set, this concern is less valid when the size of the test set is large relative to the "complexity" of hyperparameter space (as any generalization bound will attest). Note that our experiments varied only three hyperparameters over a large data set."

looks strange. You just proved that my concern was highly relevant.

If I look at your results now, I see no significant improvement by the added regularization which contradicts the main message of the paper (differences are in the 1-2 std range, completely compatible by statistical fluctuation). On the other hand, your results using AB.MR and AB.MR-L1 are really good, you do something in your tree-building procedure which seems to work better than what other people are doing (unfortunately, you don't give details on your procedure).

I'm sympathizing with you in the sense that I know that simply describing a state-of-the-art algorithm without any theoretical results will never be accepted at NIPS, but still, at this point your theoretical results are irrelevant for a practitioner (it's not your theory that makes the algorithm good), and your practical results are irrelevant for a theoretician (for the same reason: it's not your practical results that make the theory interesting or relevant). It's two papers in one with a false conclusion. I'd be happy to accept the paper if you were honest about it.
Summary: I like the idea and algorithm itself, but considering the current state of the experimental setup the decision should be a clear rejection.

Submitted by Assigned_Reviewer_11

Summary:
The authors proposes a multi-class extension of Deep Boosting framework of Cortes and Mohri. First, the authors prove a generalization bound of linear combination of base hypotheses where each base hypotheses belong to
different sets with different Radmacher complexities. The bound improves the standard multi-class generalization bound of Koltchinskii and Panchenko. Then, motivated by the generalization bound, the authors formulate an optimization problem which minimizes an upper bound of the generalization error. The proposed algorithm is a coordinate-descent style algorithm solving the optimization problem. The experimental results show that the proposed algorithm with decision trees as base hypotheses outperforms standard multi-class boosting algorithms such as AdaBoost.MR and its variants.

Comments:

The optimization problem is well motivated by the improved generalization bound of the authors. Also, the formulation naturally implies an coordinate-descent algorithm to solve it. Interestingly enough, the derived criterion to choose a weak hypothesis is the sum of the weighted error and the complexity of the hypothesis class.

The experimental result seems to show the effectiveness of the proposed method as well. But, it would be better, as done in the previous ICML'14 paper on Deep Boosting, to use validation sets to optimize parameters.

Basically, the formulation is "primal", i.e, minimizing losses (say exponential or logistic loss), similar to AdaBoost or LogitBoost.
Another formulation is based on "dual" view of boosting. For example, AdaBoost is motivated by minimizing the relative entropy from the last distribution on the sample under some linear constraints (see, Kivinen and Warmuth COLT99). Further investigations following the dual view are found in TotalBoost(Warmuth et al. ICML06), SoftBoost(Warmuth et al. NIPS07) and ERLPBoost (Warmuth et al. ALT08). The dual view for the proposed algorithm might be interesting and it might deepen the understanding of the algorithm.

After viewing the authors's comments:
Since the authors report the new experimental results based on reviewers' suggestion, I would raise my evaluation.
Summary: I think the theoretical contribution is sufficient enough for the community of NIPS.

Submitted by Assigned_Reviewer_24

This paper presents a multi-class extension to the recently proposed ensemble learning algorithm called DeepBoost. It provides new data-dependent bounds for convex ensembles in multi-class classification setup.

The paper reads well and technical derivations are correct Further it provides data-dependent bounds which are tighter than existing bounds due to the explicit dependency on the mixture weights in the convex combination. The special case of the bound derived in theorem 1, also leads to an important result as leads to a linear dependency on the number of classes. It improves the existing bounds from Koltchinskii and Panchenko, wherein this dependency is quadratic. In this sense, the bound is more useful, particularly for large number of classes. The authors also design optimization objectives and multi-class DeepBoosting algorithms which give good performance on UCI datasets as compared to the multi-class version of Adaboost and logistic regression.

However some parts of the paper need to be expanded and clarified especially the relation to the existing paper Deep Boosting in ICML, 2014 which makes the proposed approach somehow incremental as well as the experimental setup. Indeed, the proof techniques, design of the objective function, and developed algorithms are very similar in flavor to this existing work. This limits the novelty of this current work to certain extent. Another point which was not clear was that authors say in Line 143 (in section 3.1) that generalization error of $f$ and $f/\rho$ is same. In my opinion, $\rho$ has the interpretation of margin from definitions developed in equation (2), and hence it is not immediately clear why these two will admit the same generalization error. Moreover, it seems that the authors are using the labels on the test set to find the hyper-parameters of the algorithm. In the ICML paper on binary classification, the experimental setting does look different, in that paper, there is a separate validation set on which the parameter values are chosen, and not directly on the test set. The authors should clarify on this point as well.
Summary: +The paper provides data-dependent bounds which are tighter than existing bounds due to the explicit dependency on the mixture weights in the convex combination.
+ The special case of the bound derived in theorem 1, also leads to an important result as leads to a linear dependency on the number of classes.
+ The authors also design optimization objectives and multi-class DeepBoosting algorithms which give good performance on UCI datasets
- This work is somewhat incremental on the existing paper Deep Boosting in ICML, 2014
- Some points need to be clarified (experimental setup, clear novelty with respect to Deep Boosting paper in ICML, 2014)
Author Feedback
Author rebuttal: We thank the reviewers for carefully reading our paper and for their helpful comments.

- All reviewers expressed concern that the hyperparameters in our experiments were chosen by optimizing them on the test set instead of by cross-validation. We have two responses:

1) While it is of course possible to overestimate the performance of a learning algorithm by optimizing hyperparameters on the test set, this concern is less valid when the size of the test set is large relative to the "complexity" of hyperparameter space (as any generalization bound will attest). Note that our experiments varied only three hyperparameters over a large data set.

2) We agree, however, that choosing hyperparameters by cross-validation is ideal, so per Reviewer 5's suggestion we re-ran our experiments. The only change we made to the experimental setup was to divide each data set into 4 folds instead of 10: two for training, one for cross-validation, and one for testing. Note that, for instance, if we want to optimize three hyperparameters on 15x15x15 grid with 10 folds then this results in over 30000 runs of our algorithm and to make sure that we could meet the rebuttal deadline we had to reduce the number of folds. The table below reports the average and standard deviation of the error on the test folds ("pm" stands for "plus-minus"). Observe that MDeepBoost has the lowest average test error on every data set (except for one data set where it is tied with AdaBoost.MR-L1); in fact we observed that MDeepBoost had the lowest test error on almost every fold.

pendigits
AdaBoost.MR-------0.014 pm 0.0025
AdaBoost.MR-L1---0.014 pm 0.0013
MDeepBoost--------0.012 pm 0.0011

statlog
AdaBoost.MR-------0.029 pm 0.0026
AdaBoost.MR-L1---0.026 pm 0.0071
MDeepBoost--------0.024 pm 0.0008

handwritten
AdaBoost.MR-------0.024 pm 0.0011
AdaBoost.MR-L1---0.025 pm 0.0018
MDeepBoost--------0.021 pm 0.0015

letters
AdaBoost.MR-------0.065 pm 0.0018
AdaBoost.MR-L1---0.059 pm 0.0059
MDeepBoost--------0.058 pm 0.0039

yeast
AdaBoost.MR-------0.415 pm 0.0353
AdaBoost.MR-L1---0.410 pm 0.0324
MDeepBoost--------0.407 pm 0.0282

abalone
AdaBoost.MR-------0.739 pm 0.0016
AdaBoost.MR-L1---0.737 pm 0.0065
MDeepBoost--------0.735 pm 0.0045

satimage
AdaBoost.MR-------0.120 pm 0.0123
AdaBoost.MR-L1---0.117 pm 0.0096
MDeepBoost--------0.117 pm 0.0087

pageblocks
AdaBoost.MR-------0.035 pm 0.0045
AdaBoost.MR-L1---0.035 pm 0.0031
MDeepBoost--------0.033 pm 0.0014

Finally, we would like to add that it was mainly because of the lack of time that we reported results for parameters tuned on test data instead of cross-validation results. We were able to obtain new results quicker both due to the reduced number of folds and a new improved implementation. Should our work be accepted, we will include the newly presented cross-validation experiments and modify conclusions that we make in the paper to reflect these results.

- Reviewer 5 objects to the title of the paper. While we agree that the title may lead a reader to incorrectly associate our paper with the literature on neural networks, the paper by Cortes, Mohri and Syed (2014), which our work extends, also used "deep" in the title, so we feel that changing the title of our paper would be even more confusing than keeping it as it is.

- Reviewer 5 has helpfully provided several additional references, which we would be happy to cite in the camera-ready version. We will also clarify the ambiguous notation in the caption.

- Reviewer 5 would like us to provide an open source version of our algorithm. In fact our implementation was based on an open source version of the algorithm from Cortes, Mohri and Syed (2014), and if our paper is accepted we can ask them whether they would be willing to include our code in their repository.

- We agree with Reviewer 11 that the dual view provides an alternative way of justifying the algorithm and that exploring this direction would be interesting.

- Reviewer 24 felt that our contribution was incremental relative to Cortes, Mohri and Syed (2014). We wish to emphasize that our paper contains many novel results, including new multi-class generalization bounds, analysis of and comparison several objectives for the multi-class setting, new algorithms for each of these objectives, consistency proofs, convergence proofs, and experimental results in the multi-class setting.

- Reviewer 24 asked us to clarify why generalization error of $f$ and $f/\rho$ is same in Line 143 (in section 3.1). Please recall that multiplying $f$ by a positive constant does not affect the way a point is classified since the class with the highest score is selected. Therefore the zero-one loss of f is invariant to rescaling f by a positive constant.

In conclusion, we would like to emphasize again the theoretical and algorithmic contributions of our work. We give new data-dependent bounds for convex ensembles in the multi-class classification setting. These bounds are finer than existing bounds for multi-class classification both thanks to a linear (instead of quadratic) dependency on the number of classes and, more importantly, by virtue of a more favorable complexity term expressed as an average of the Rademacher complexities based on the ensemble's mixture weights. Based on these guarantees, we introduce several new multi-class ensemble algorithms and establish the H-consistency and convergence results for several of these algorithms.